# Mechanical stretch induces hair regeneration through the alternative activation of macrophages

Szu-Ying Chu[1,2,3], Chih-Hung Chou [4], Hsien-Da Huang[5], Meng-Hua Yen[6], Hsiao-Chin Hong[7], Po-Han Chao[2], Yu-Hsuan Wang[8,9], Po-Yu Chen[1,3], Shi-Xin Nian[1,3], Yu-Ru Chen [2], Li-Ying Liou[1,3], Yu-Chen Liu[7], Hui-Mei Chen[7], Feng-Mao Lin[7], Yun-Ting Chang[1,3], Chih-Chiang Chen[1,2,3] & Oscar K. Lee[2,8,9]

Tissues and cells in organism are continuously exposed to complex mechanical cues from the environment. Mechanical stimulations affect cell proliferation, differentiation, and migration, as well as determining tissue homeostasis and repair. By using a specially designed skin-stretching device, we discover that hair stem cells proliferate in response to stretch and hair regeneration occurs only when applying proper strain for an appropriate duration. A counterbalance between WNT and BMP-2 and the subsequent two-step mechanism are identified through molecular and genetic analyses. Macrophages are first recruited by chemokines produced by stretch and polarized to M2 phenotype. Growth factors such as HGF and IGF-1, released by M2 macrophages, then activate stem cells and facilitate hair regeneration. A hierarchical control system is revealed, from mechanical and chemical signals to cell behaviors and tissue responses, elucidating avenues of regenerative medicine and disease control by demonstrating the potential to manipulate cellular processes through simple mechanical stimulation.

[1] Department of Dermatology, Taipei Veterans General Hospital, Taipei 112, Taiwan. [2] Institute of Clinical Medicine, National Yang-Ming University, Taipei 112, Taiwan. [3] Department of Dermatology, National Yang-Ming University, Taipei 112, Taiwan. [4] Department of Biological Science and Technology, Center for Intelligent Drug Systems and Smart Bio-devices (IDS2B), National Chiao Tung University, Hsinchu 300, Taiwan. [5] Warshel Institute for Computational Biology, School of Life and Health Sciences, School of Sciences and Engineering, The Chinese University of Hong Kong, Shenzhen 518172, China. [6] Department of Electronic Engineering, National Chin-Yi University of Technology, Taichung 411, Taiwan. [7] Department of Biological Science and Technology, Institute of Bioinformatics and Systems Biology, National Chiao Tung University, Hsinchu 300, Taiwan. [8] Department of Orthopaedics and Traumatology, The Chinese University of Hong Kong, Hong Kong 999077, China. [9] Institute for Tissue Engineering and Regenerative Medicine, The Chinese University of Hong Kong, Hong Kong 999077, China. Correspondence and requests for materials should be addressed to C.-C.C. (email: docs1.tw@yahoo.com.tw) or to O.K.L. (email: oscarlee9203@gmail.com)

Life relies on key cellular processes such as cell proliferation, differentiation, and migration. These processes are influenced by several factors and interactions that require robust control for safe and effective tissue regeneration[1,2]. Major emphasis has been placed on chemical gradients to explain how microenvironments direct cell behavior to achieve functional tissue regeneration[1,3,4]. This approach, however, cannot account for remarkable examples of the self-organization of cells to form organoids in culture saturated with chemical factors[5]. The prominence of mechanical forces in the initiation and maintenance of 3-D tissue architecture has been overlooked in previous studies[6]. In fact, research has indicated that cell proliferation and differentiation can be induced not only by chemical factors but also by mechanical stimulation, which initiates upstream signals to activate downstream chemical signaling[3,7]. Therefore, one challenge in the fields of regenerative medicine and disease control is understanding how forces orchestrate chemical signals to initiate or sustain biological patterns in the tissues.

Mammalian hairs constitute an ideal model to address these topics. Their visibility and accessibility make recording their spatiotemporal patterns feasible. Furthermore, because hair follicles undergo a continuous cycle throughout life, they provide an excellent platform for evaluating regeneration potential in response to mechanical stimulation. Another study we conducted demonstrated that hair plucking can induce an efficient regeneration response[8], which suggests that the mechanical force elicited by plucking may play key roles in regeneration process. Furthermore, research has confirmed that arrector pili muscles, which connect the bulge region of a hair follicle (where the hair stem cells reside) to the basement membrane and perform the functions for goosebumps formation and hair follicle alignment through muscle contracture, are involved in the hair regeneration cycle[9–11]. These phenomena prompted our interest in whether and how mechanical force influences stem cell behavior in vivo.

To fulfill this research purpose, we designed a specialized skin-stretching device that can identify how mechanical forces affect hair regeneration by modifying the strain. We discover that mechanical stretch activates hair stem cells and facilitates hair growth under specific strain and duration conditions. Combined with the results of molecular analyses, as well as genetic and pharmacological manipulation, we discover that stretch stimulates chemokine production and macrophage recruitment. These macrophages undergo M2 polarization and produce various growth factors such as hepatocyte growth factor (HGF) and insulin-like growth factor 1 (IGF-1) to induce hair regeneration.

## Results

**Stretch-induced regeneration is strain/duration-dependent**. To evaluate how mechanical stretch affect hair follicles, a specially designed skin-stretching device was used that can be fixed on back skin to create programmable stretch (Supplementary Fig. 1a). Eight-week-old mice were used to observe alteration to the hair cycle in response to stretch when their hairs were maintained in the physiological telogen phase. Two specific variables, strain and stretch duration, were controlled to parameterize the efficacy of anagen initiation and hair growth. In the first set of experiments, skin was stretched for 7 days with different initial strains (0, 20, 33, and 40%) (Fig. 1a, Supplementary Fig. 2a, b). When strain reached 33% or greater, hair regeneration was induced within the whole stretching area. By contrast, hair follicles remained in a telogen or resting phase when the strain was less than 20%. These results demonstrated the vital role of strain in controlling hair cycle.

We next determined whether stretch duration affects the activation of hair stem cells. The same 33% initial strain

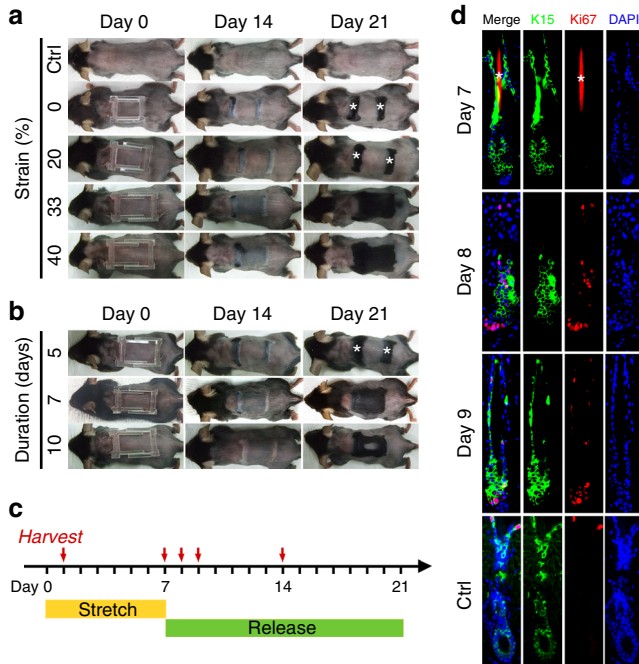

**Fig. 1** Stretch-induced hair regeneration is dependent on strain and duration. **a** Stretch for 7 days under different amounts of strain. Note hair regeneration occurred in response to 33 or 40% strain, but not in the control (without the device) or under 0 or 20% strain; $n = 6$ for each group. Day 0 represents the day on which strain was applied. *Anagen initiated in glue-fixed area due to hair plucking when removing the skin-stretching device. **b** Stretch under 33% strain for different durations. Note hair regeneration occurred in the 7-day (total) and 10-day (only in peripheral area) duration groups, but not in the 5-day duration group; $n = 6$ for each group. **c** Schematic of the optimal stretching conditions and sample collection times. **d** Dual immunostaining for K15 and Ki67 revealed that K15$^+$ hair stem cells began to proliferate on days 8 and 9 when stretch was released. Scale bar = 50 μm. *Autofluorescence of hair shafts

was applied for different durations (5, 7, and 10 days), and three different responses were observed (Fig. 1b, Supplementary Fig. 2a, c). When the strain was applied for only 5 days, no hair regeneration occurred. When the stretch duration was 10 days, hair growth was only observed in peripheral region. However, when strain was applied for 7 days, anagen initiation and hair growth were observed evenly in the whole stretching area. Accordingly, this stretching condition (33% strain with 7-day duration) was determined to be the optimal condition for inducing hair regeneration (Fig. 1c). To further validate this finding, the optimal parameter was applied during the synchronized telogen period[12]. In other words, whole back skin was depilated to induce simultaneous anagen initiation. After full development of the anagen phase, the skin entered a homogeneous status, or synchronized telogen phase, and was then stretched under this condition. Notably, hair regeneration occurred as efficiently as in mice whose skin was stretched during the physiologic telogen phase (Supplementary Fig. 2d). Considered together, these results suggest that strain and duration cooperatively regulate telogen-to-anagen transition.

Hair cycle is tightly controlled by signals from both the intrafollicular microenvironment and surrounding macroenvironment[13–16]. Among these signals, WNT/β-catenin and bone morphogenetic protein (BMP) are an activator and inhibitor, respectively, that are critical to the hair cycle[14,15,17–19]. To evaluate alteration of these hair cycle regulators in response to stretch, we examined the expression pattern of WNT and BMP

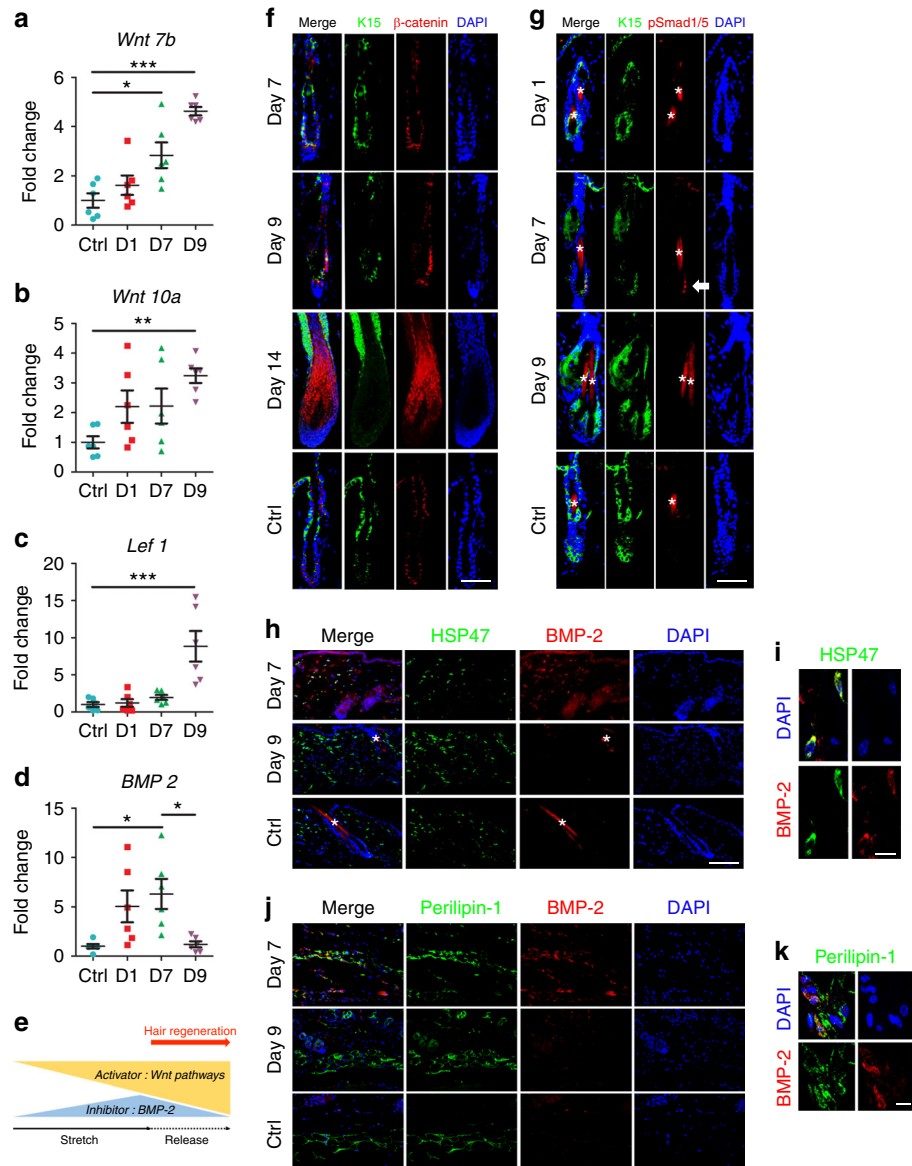

**Fig. 2** WNT and BMP signaling pathways play crucial roles in regeneration. **a–d** Real-time PCR for WNT and BMP signals, including *Wnt7b* (**a**), *Wnt10a* (**b**), *Lef1* (**c**), and *Bmp2* (**d**) in response to stretch (day 1 and day 7) and release of stretch (day 9); *n* = 6 for each group. **e** Schematic of hair cycle activator and inhibitor in response to strain alteration. **f** Dual immunostaining for K15 and β-catenin revealed nuclear staining of β-catenin in hair follicles at day 9 and day 14 when stretch was released. Scale bar = 50 μm. **g** Dual immunostaining for K15 and phospho-Smad1/5 (pSmad1/5) revealed the nuclear staining of pSmad1/5 in hair stem cells at day 7 when skin was stretched (arrow). Scale bar = 50 μm. **h, i** Immunostaining (**h**) and confocal microscope (**i**) revealed co-localization of HSP47 and BMP-2 signals at day 7 when skin was stretched. Scale bar = 100 μm (**h**) or 10 μm (**i**). **j, k** Immunostaining (**j**) and confocal microscope (**k**) revealed co-localization of Perilipin-1 and BMP-2 signals at day 7 when skin was stretched. Scale bar = 100 μm (**j**) or 10 μm (**k**). Statistical significance was determined using ANOVA followed by a Bonferroni post hoc test. Data are presented as means ± SEM. *$p < 0.05$. **$p < 0.01$. ***$p < 0.001$. Source data are provided as a Source Data file. *Autofluorescence of hair shafts in **g**, **h**

signals under the optimal stretching condition (33% strain with 7-day duration). *Wnt7b*, *Wnt10a*, and *Lef1* (a nuclear responder of *Wnt* signals) peaked at day 9 when stretch was released (Fig. 2a-c). Immunostaining revealed extensive nuclear β-catenin expression in hair matrix, which demonstrated the effects of WNT signals on hair follicles (Fig. 2f). The hair cycle inhibitor *Bmp2* was also upregulated in response to stretch from day 1 to day 7 (Fig. 2d). Notably, the *Bmp2* level was lower at day 9 when stretch was released (Fig. 2d). Immunostaining revealed that BMP-2 was expressed by both dermal fibroblasts and adipocytes (Fig. 2h–k). Furthermore, nuclear pSMAD1/5 expression was

observed in hair stem cells at day 7, which confirmed that BMP-2 affected hair stem cells (Fig. 2g). Finally, we conducted dual immunofluorescence staining of K15 and Ki67 to evaluate the activity of hair stem cells. Conspicuous proliferation of hair stem cells was observed at day 8 and 9 when stretch was released (Fig. 1d). Few apoptotic cells were identified in the stretched skin, indicating that stretch-induced hair regeneration did not result from cell damage caused by our stretching device or the stretching procedure (Supplementary Fig. 3).

Taken together, these results indicate that mechanical stretch can induce hair regeneration in a threshold-dependent manner.

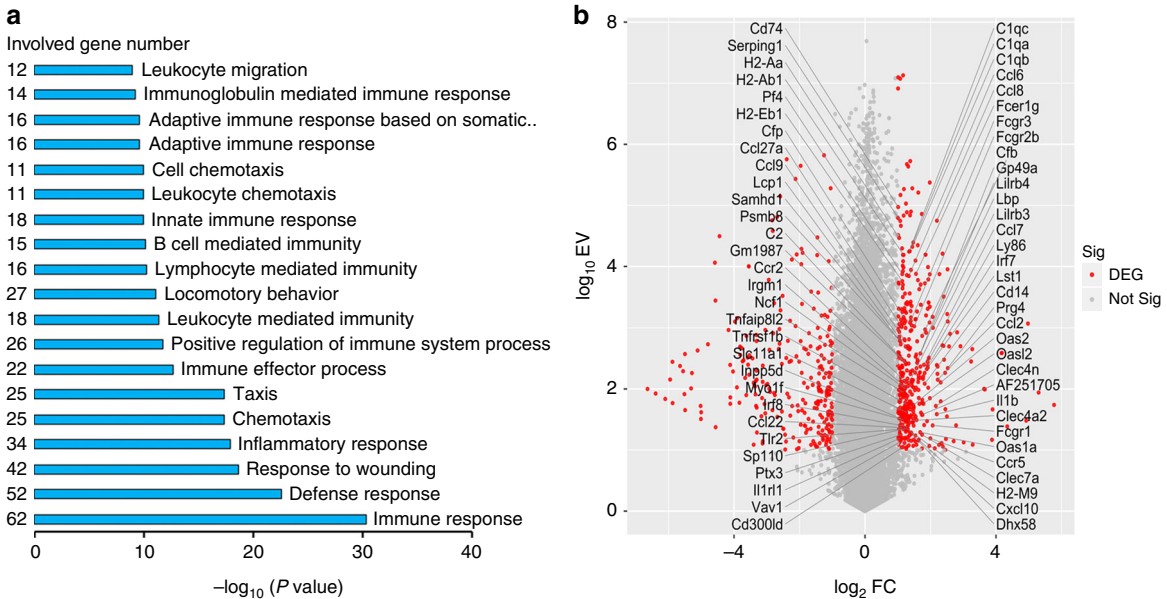

**Fig. 3** Functional category enrichment analysis in response to stretch. **a** Top-ranked upregulated biological processes revealed through a gene ontology analysis, mostly related to inflammatory processes. **b** Volcano plot of differentially expressed genes (DEGs) in immune response gene ontology (GO:0006955). Red points represent DEGs with an expression value ≥10 and fold change ≥2 or ≤0.5

The counterbalance between the activator, WNT signaling pathway, as well as the inhibitor, BMP-2 plays the most important role in determining whether regeneration occurs (Fig. 2e).

**Stretch elicits macrophage-predominated inflammation**. To further evaluate the possible mechanisms instrumental in triggering hair stem cell activation, we performed RNA-sequencing (RNA-seq) and grouped genes into their respective categories based on Gene Ontology (GO) in stretch-stimulated and control mice. We identified 416 upregulated genes and 264 down-regulated genes from all 22,579 genes. Notably, GO term analysis of the upregulated genes revealed that most of them were associated with inflammatory processes such as immune response, defense response, chemotaxis, immune effector process, or leukocyte mediated immunity (Fig. 3a, Supplementary Table 1). The upregulated genes in immune response GO (GO:0006955) are presented in Fig. 3b.

To better understand the inflammatory process in response to mechanical stretch, immunostaining was performed to characterize the types and spatiotemporal distribution of inflammatory cells under the optimal stretching condition for effectively inducing hair regeneration (33% strain with 7-day duration) (Fig. 4a). At day 1, the dermis and hypodermal adipose layer were heavily infiltrated with F4/80-positive macrophages. Macrophage infiltration persisted from day 1 to day 7 when stretch was applied, and it was still observed at day 9 when stretch was released. Flow cytometry was conducted to quantify the numbers of CD45$^+$F4/80$^+$ macrophages, which revealed that the percentage of macrophages increased when the skin was stretched (Fig. 4b). Notably, the increment of macrophages was maintained after mechanical stimulation was removed on day 9 (Fig. 4b), implying that inflammatory process was initiated in response to stretching and sustained after the release of stretch.

To evaluate whether macrophages play a functional role in stretch-induced hair regeneration, we used a chemical inhibitor to ablate them (Fig. 4d). Application of clodronate liposome significantly decreased the number of macrophages (Fig. 4c, e) and the suppression of macrophage function was demonstrated to

prohibit hair regeneration within the stretching area in vivo (Fig. 4f). This result confirmed the indispensability of macrophages in stretch-induced hair regeneration.

**Stretch activates chemokines for macrophage recruitment**. To identify the chemotactic signals that attract and recruit macrophages to stretched skin, we analyzed the most altered chemokines revealed in RNA-seq through real-time polymerase chain reaction (RT-PCR). We discovered that *Ccl2*, *Ccl3*, *Ccl6*, *Ccl7*, *Ccl12*, and *Ccl22* were highly upregulated after stretching but peaked at different times (Fig. 4g). Among the upregulated chemokines, CCL2 has been reported to respond to mechanical stretch in various cell types[20,21]. CCL2 is also one of the major chemoattractants for macrophages with significant functional roles in attracting macrophages to inflamed skin[8,22]. To evaluate whether CCL2 is essential for hair regeneration under stretch conditions, we applied 33% initial strain to stretch the skin of CCL2 null mice for 7 days. Notably, knockout of CCL2 failed to prevent anagen initiation, and hair regenerated as efficient as it would in wild-type mice (Fig. 1a, b, Supplementary Fig. 4a). Immunofluorescence of F4/80 still revealed heavily infiltrated macrophages in stretched skin (Supplementary Fig. 4b), implying that, in addition to CCL2, various chemokines were also involved in macrophage recruitment in stretched skin. Considered together, these findings demonstrate that mechanical stretch stimulates the production of multiple chemokines that facilitated macrophage recruitment.

**M2 macrophages respond to strain and induce regeneration**. Macrophages in higher animals undergo classical activation (M1 polarization) or alternative activation (M2 polarization) in response to environmental signals[23,24]. M1 macrophages execute kill function, whereas M2 macrophages are responsible for repair[25,26]. The unique abilities of different macrophages are essential for maintaining tissue homeostasis[25]. To acquire further insights into if and how macrophages polarize in response to stretch, percentages of M1 (CD45$^+$F4/80$^+$CD86$^+$CD206$^-$) and M2 (CD45$^+$F4/80$^+$CD86$^-$CD206$^+$) macrophages were examined using flow cytometry under the optimal stretching condition

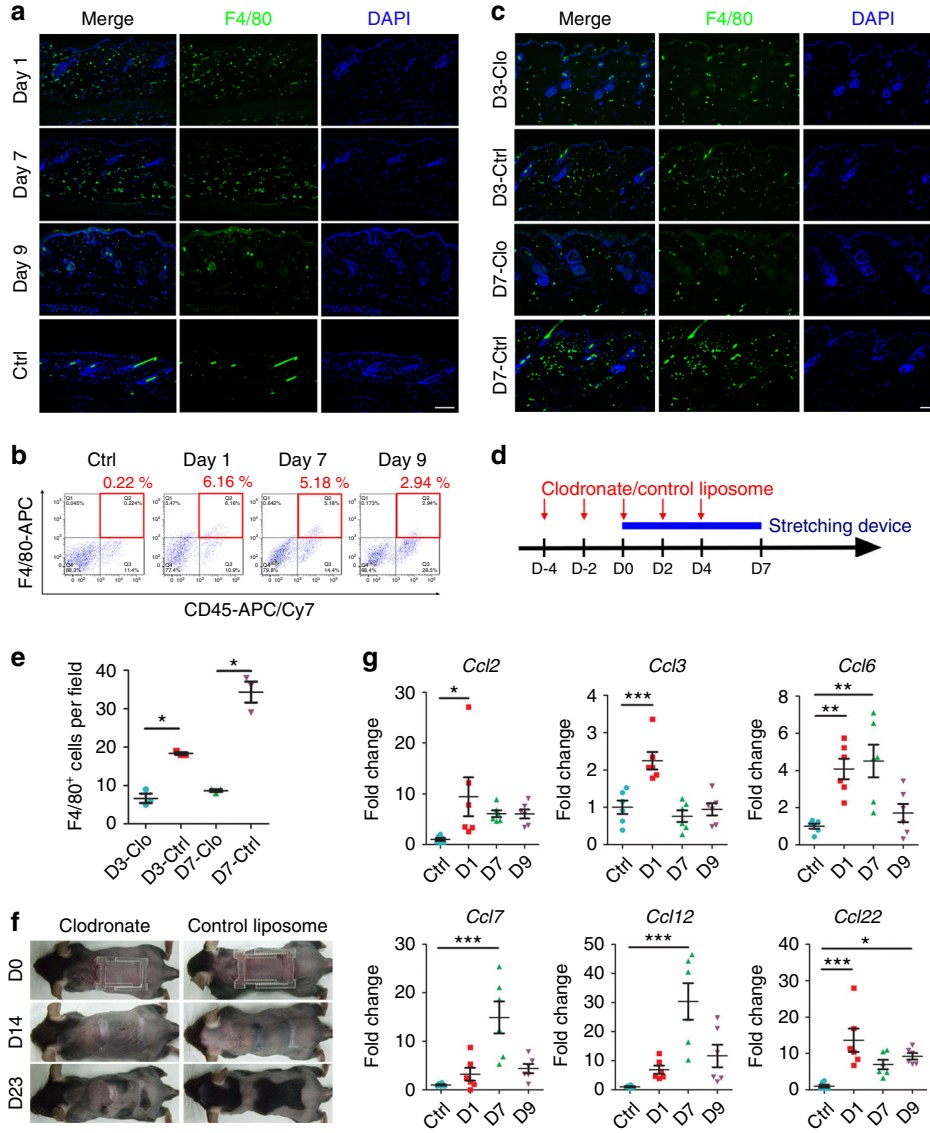

**Fig. 4** Macrophages are major mediators in stretch-induced hair regeneration. **a** Immunostaining of F4/80 indicated extensive macrophage infiltration in response to stretch (day 1 and day 7) that was sustained after stretch was released (day 9). **b** Representative FACS scatterplots display the percentages of CD45$^+$F4/80$^+$ cells activated in response to stretch (day 1 and day 7) and release of stretch (day 9). **c** Immunostaining of F4/80 revealed decreased macrophage infiltration after skin was stretched for 3 days and 7 days in the clodronate injection (Clo) group compared with the control liposome injection (Ctrl) group. **d** Schematic of subcutaneous injection of clodronate liposomes to target macrophages. **e** Quantification of F4/80$^+$ cells in the Clo and Ctrl groups; $n = 3$ for each group. **f** Stretch-induced hair regeneration is halted by clodronate injection; $n = 6$ for each group. **g** Real-time PCR for *Ccl2*, *Ccl3*, *Ccl6*, *Ccl7*, *Ccl12*, and *Ccl22* in response to stretch (day 1 and day 7) and release of stretch (day 9); $n = 6$ for each group. Statistical significance was determined using Student's two-tailed *t*-test (**e**) or ANOVA followed by a Bonferroni post hoc test (**g**). Data are presented as means ± SEM. *$p < 0.05$. **$p < 0.01$. ***$p < 0.001$. Scale bars = 100 μm. Source data are provided as a Source Data file

(33% strain with 7-day duration). Before stretch, all of the resident macrophages were unpolarized (CD45$^+$F4/80$^+$CD86$^-$CD206$^-$); the percentage of M2 macrophages gradually increased in response to stretch from day 1 (1.87%) to day 7 (12.4%) (Fig. 5a, b). More importantly, M2 macrophages were further increased at day 9 (29.4%) when stretch was released (Fig. 5a, b). To more accurately detect M2 macrophages, we sorted CD45$^+$F4/80$^+$ macrophages and arranged RT-PCR for *Arginase1* and *Ym1* (M2 markers). Both *Arginase1* and *Ym1* increased in response to stretch (day 7) compared with non-stretched skin (Fig. 5c, d). The level of *Ym1* further increased when stretch was released (day 9). Dual immunostaining of F4/80 and Arginase also revealed that the increment of the M2 subtype was more conspicuous after the stretch had been released

(Fig. 5g). Because M2 polarization is mediated by cytokines such as IL-4[26,27], we examined the expression of IL-4 in stretched skin. Notably, *Il4* was marked as upregulated soon after stretching (Fig. 5e). Collectively, these results demonstrated that macrophage subtype was skewed toward the M2 phenotype by altering the strain.

Our observation thus suggested that the strain-stimulated macroenvironment recruited macrophages through chemokine release and polarized them into M2 subtypes. We therefore hypothesized that M2 macrophages are the key mediators in stretch-induced hair regeneration. To test this hypothesis, we injected clodronate encapsulated mannosylated liposome into mice skin to specifically deplete M2 macrophages (Fig. 5h)[28]. Hair regeneration was noticeably perturbed by M2 depletion

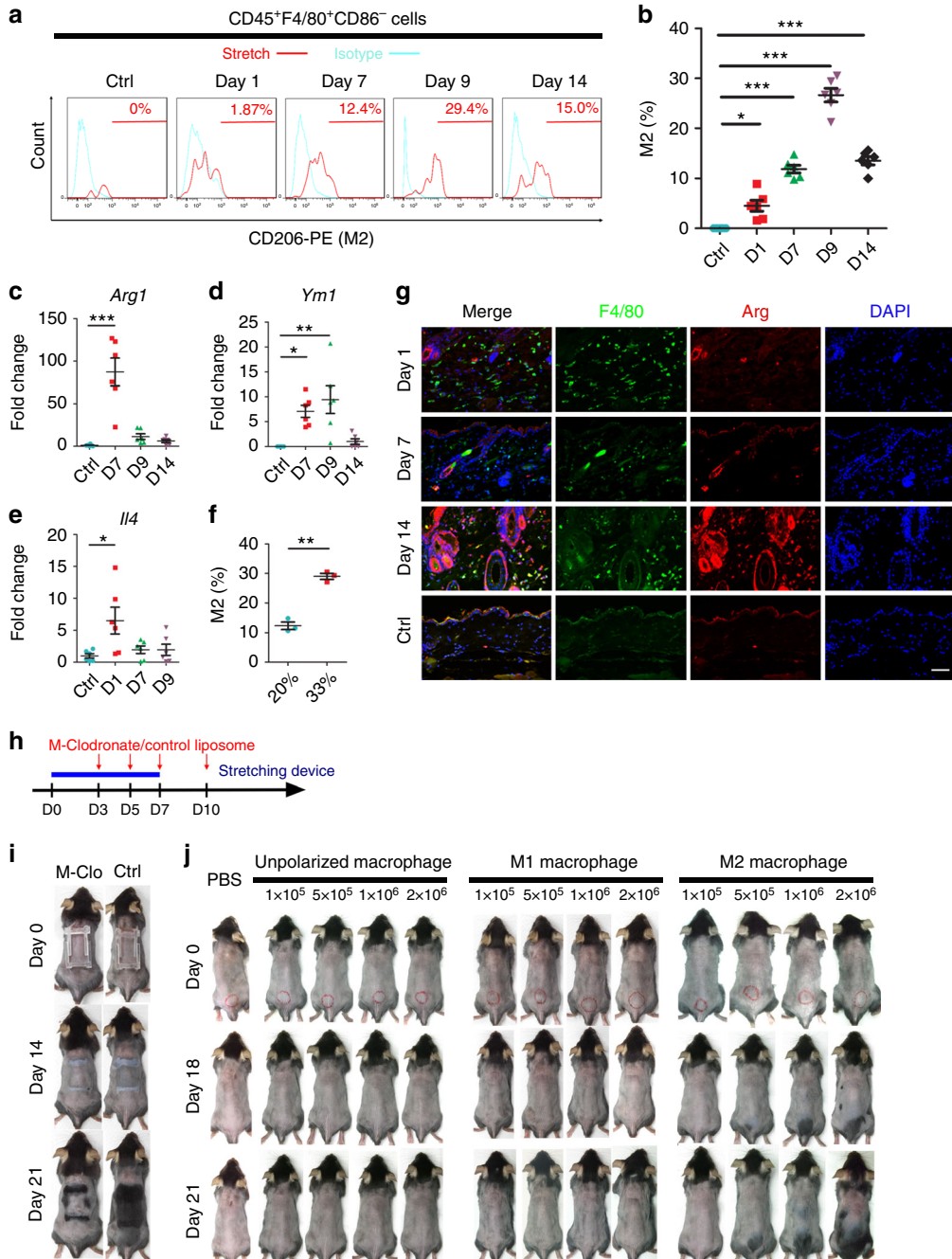

**Fig. 5** M2 polarization induced by strain alteration is crucial for regeneration. **a**, **b** Representative FACS histograms (**a**) and quantification (**b**) of the percentages of CD45+F4/80+CD206+CD86- M2 macrophages activated in response to stretch (day 1 and day 7) and release of stretch (day 9 and day 14); $n = 6$ for each group. **c–e** Real-time PCR for Arginase-1 (Arg1) (**c**), Ym1 (**d**), and Il4 (**e**) in response to stretch (day 1 and day 7) and release of stretch (day 9 and day 14); $n = 6$ for each group. **f** Quantification using flow cytometry of M2 macrophages in response to 20 or 33% strain; $n = 3$ for each group. **g** Dual immunostaining of F4/80 and Arginase indicated that double positive M2 macrophages were more conspicuous at day 14. **h** Schematic of subcutaneous injection with mannosylated clodronate liposomes (M-Clodronate) to deplete M2 macrophages. **i** Stretch-induced hair regeneration was perturbed by mannosylated clodronate liposomes (M-Clo) injection; $n = 6$ for each group. **j** Transplantation of unpolarized and polarized M1 or M2 macrophages into the back skin of mice during the telogen phase. Note only M2 macrophages caused hair regeneration, and the intensity of hair growth was proportional to the number of transplanted cells. The red circle at day 0 indicates the injection area; $n = 6$ for each group. Statistical significance was determined by conducting ANOVA followed by a Bonferroni post hoc test (**b–e**) or Student's two-tailed $t$-test (**f**). Data are presented as means ± SEM. *$p < 0.05$. **$p < 0.01$. ***$p < 0.001$. Scale bar = 50 μm. Source data are provided as a Source Data file

(Fig. 5i). To conclusively determine whether M2 macrophages induce hair regeneration, we polarized bone marrow-derived macrophages (BMDMs) into M1 or M2 subtypes and performed subcutaneous transplantation during the telogen phase to observe the alteration in the hair cycle (Fig. 5j). Strikingly, hair regeneration was more substantially induced by transplantation of M2 than by transplantation of unpolarized or polarized M1 macrophages. The intensity of anagen initiation and hair growth was proportional to the number of transplanted M2 macrophages (Fig. 5j). Finally, we compared the number of M2 macrophages in

the regenerated stretching condition (33% strain for 7 days) and nonregenerated stretching condition (20% strain for 7 days). The percentage of M2 macrophages was significantly increased under 33% strain compared with 20% strain (Fig. 5f). These findings provide compelling evidence suggesting that macrophage infiltration alone was insufficient for eliciting hair growth and that the alternative activation of macrophages (M2 polarization) was essential for hair stem cell activation.

**M2 macrophages facilitate regeneration with growth factors**. M2 macrophages are known to produce a wide array of growth factors[29,30]. To search for functional mediators secreted by macrophages, we sorted F4/80$^+$ macrophages under the optimal stretch condition and perform RT-PCR to evaluate the temporal alteration of various growth factors (Fig. 6a). Notably, growth factors such as *Hgf*, *Igf1*, and fibroblast growth factor 10 (*Fgf10*) were significantly upregulated at day 9 when stretch was being released. Fibroblast growth factor 2 (*Fgf2*), vascular endothelial growth factor (*Vegf*) and keratinocyte growth factor (*Kgf*) were not significantly increased either by stretch or release of the skin. Because M2 macrophages play an essential role in our model, we compared the expression pattern of growth factors between CD45$^+$F4/80$^+$CD206$^+$ (M2) and CD45$^+$F4/80$^+$CD206$^-$ (non-M2) cells at day 9, when the increment of growth factors was more conspicuous. *Hgf* and *Igf1* were significantly upregulated in M2 compared with non-M2 cells (Fig. 6b). The time-dependent increase of growth factors appeared to be relevant because it correlated with the timed activation of hair stem cells and the timing of telogn-to-anagen transition (Fig. 1d). To determine whether growth factors alone are sufficient for triggering hair regeneration, we injected HGF-coated or IGF-1-coated beads into the back skin of mice during telogen phase. Within 5 days following bead implantation, hair stem cells within the hair follicles adjacent to either HGF-coated or IGF1-coated beads displayed proliferation by expression of Ki67 (Fig. 6c). The increase in Ki67 labeling corresponded with that of hair germ size. This indicated that hair germs had begun to proliferate, which represented anagen initiation in response to HGF and IGF-1. Collectively, our findings revealed both the source and the functional importance of growth factors in stretch-induced hair regeneration.

## Discussion

Studies have emphasized the importance of stretch-induced regeneration[31–34]. However, most have focused on single cell population, such as keratinocytes, fibroblasts, or bone cells[34–36]; much less is known regarding the system complexity of living organism composed of multicell population. Although some studies have attempted to mimic the in vivo state by coculturing keratinocytes and fibroblasts[37,38], these systems are still far from representing a physiological condition, because the existence of skin appendages is neglected and the inflammation process is prohibited without a vascular system. Thus, how mechanical force or stretch affects stem cell behavior in multipopulation systems remains unclear.

To overcome the aforementioned challenges, a specially designed in vivo skin-stretching device was utilized in this study to elucidate the regeneration potential of hair follicles in response to stretch in an unsimplified three-layer skin system. We successfully demonstrated that mechanical stretch can efficiently induce hair regeneration under specific stretching conditions. This type of regeneration is threshold dependent; this is similar to our previous results regarding quorum sensing[8]. In accordance with our findings, expansion of the human scalp has been reported to facilitate hair growth; however, the underlying mechanism had not been investigated in a clinical study[39]. By

using molecular analyses, genetic studies, and pharmacological manipulation, we explored the hierarchical control of mechanical stretch, chemokine induction, macrophage recruitment, growth factors production, and hair regeneration.

The concept of the alternative activation of macrophages has accumulated research support, and its roles in repair and regeneration have been explored in different organ systems, such as those of the heart, kidney, and spinal cord[27,29,40–43]. M2 macrophages are therefore considered to possess therapeutic potential in treating various diseases[40,44]. However, although their spatiotemporal patterns for epithelial tissues are relative easy to observe and measure, the roles of M2 macrophages in hair regeneration remain largely unknown. Studies have demonstrated that TNF-α-activated or IL-1β-activated macrophages can facilitate hair growth in wounds[45,46], clarifying how injury and repair affected hair regeneration. However, the polarization of macrophages during wound healing is highly dynamic, and the specific roles of M2 macrophages are difficult to discern in such complicated macroenvironment[29]. In the present study, we discovered that mechanical stretch can induce alternative activation of macrophages, which have functional roles in hair regeneration (Fig. 7a, b). We further revealed that growth factors, especially HGF and IGF-1, which are produced by M2 macrophages, are functional mediators in stretch-induced hair regeneration (Fig. 7b). Although the positive effects of growth factors on hair regeneration have been reported, their high cost limits their usage in clinical practice[17,47]. Our study thus provides an approach to facilitate hair regeneration by stimulating intrinsic growth factors through the application of proper mechanical force.

Macrophages display distinct functional phenotypes in response to environmental cues such as oxygen tension and the presence of cytokines and metabolites[24,29,48]. In this study, we unveiled the potential of using mechanical stretch to polarize macrophages toward M2 phenotype in vivo, and the RNA-seq data enabled us to acquire insight into how this process occurs. In addition to the well-established stimulator IL-4, the metabolic status was markedly altered in response to stretch (Supplementary Table 2). Notably, the use of fatty acid oxidation pathways, including those for saturated and unsaturated fatty acid metabolic process (GO:0006631 and GO:0033559), were significantly increased ($p < 0.05$) while rapid glucose consumption pathways, including those for glucose metabolic process (GO:0006006), hexose metabolic process (GO:0019318), and glycolysis (mmu0010) were significantly decreased ($p < 0.05$). These findings are in consistent with the concepts that M1 favors aerobic glycolysis and M2 uses fatty acid oxidation[48,49]. Taken together, mechanical stretch might facilitate M2 polarization through regulation of inflammatory cytokines or metabolic processes though the precise mechanism requires further evaluation.

Our collective results identify the roles of chemokines in macrophage recruitment. However, the upstream signals responsible for prompting mechanical stretch to activate chemokines remain unknown. Studies have revealed that fragments of the extracellular matrix (ECM) can upregulate chemokines, promote macrophage recruitment, enhance phagocytic functions, and induce cytokines[50–53]. Furthermore, disruptions to the cell junction may also be functional signals for dendritic cell maturation[54]. Accordingly, tissue damage, such as rupture of the ECM or cell junction induced by stretch, might play a role in macrophage recruitment or polarization. Moreover, the links between force distribution, tissue damage, morphogen gradients and cell responses have acquired substantial research interest and will be clarified in the future.

In summary, our work reveals the mechanical force—immune response—regeneration process axis and demonstrates how organs achieve regeneration in response to mechanical force. We

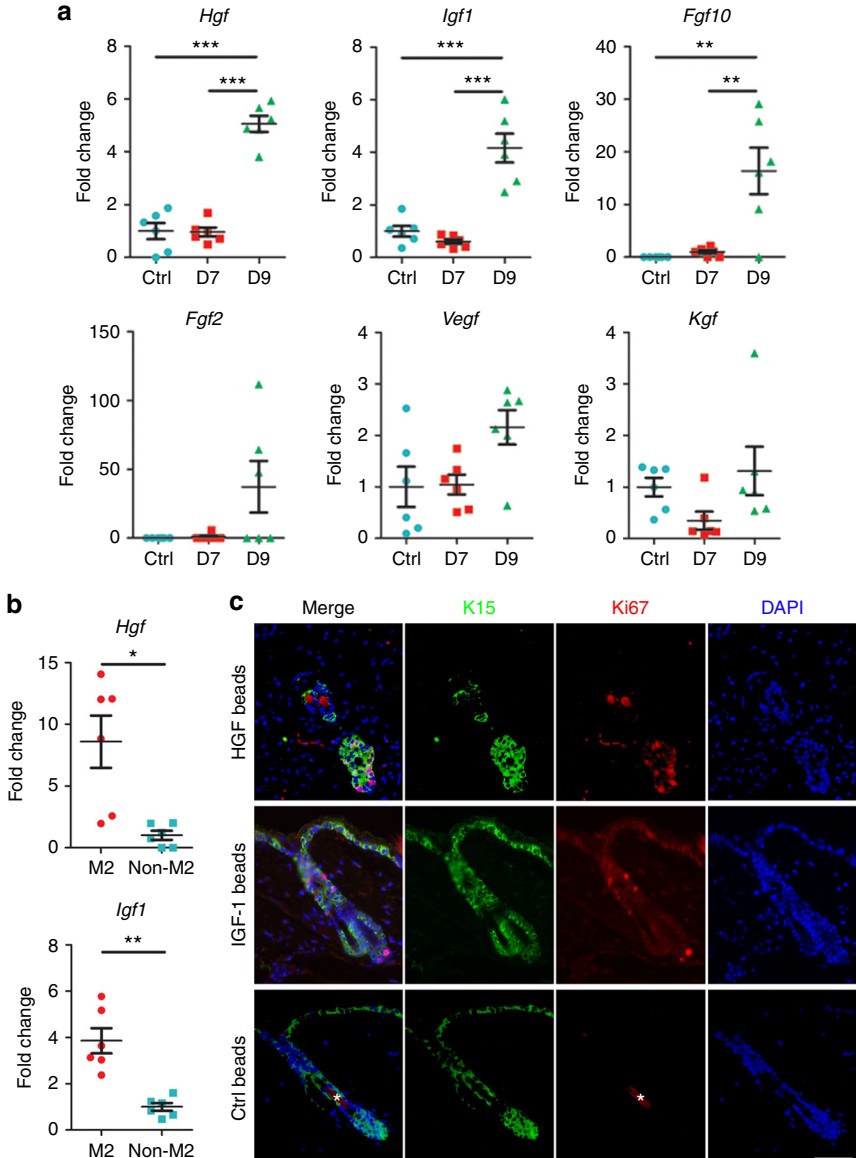

**Fig. 6** M2 macrophages produce growth factors to facilitate hair regeneration. **a** Real-time PCR of the sorted macrophages revealed the kinetics of gene expression of various growth factors in response to stretch (day 7) and release of stretch (day 9); $n = 6$ for each group. **b** Expression levels of growth factors within M2 (CD45+F4/80+CD206+) and non-M2 (CD45+F4/80+CD206-) macrophages according to real-time PCR; $n = 6$ for each group. **c** Dual immunostaining of K15 and Ki67 demonstrated that the injection of HGF-coated or IGF-1- coated beads during the telogen phase precociously activates K15 + hair stem cells. *Autofluorescence of hair shafts. Statistical significance was determined by conducting ANOVA followed by a Bonferroni post hoc test (**a**) or Student's two-tailed $t$-test (**b**). Data are presented as means ± SEM. *$p < 0.05$. **$p < 0.01$. ***$p < 0.001$. Scale bar = 50 μm. Source data are provided as a Source Data file

believe that this process presents in the regeneration of tissue and organs beyond the skin and it points to a different direction for the future of regeneration medicine. However, more research is required to clarify how mechanical stimulation modulates growth factors secretion and whether other physiologic pathways (e.g., mechano-transduction pathways) participate in stretch-induced hair regeneration.

## Methods

**Animal procedures**. C57BL/6 mice were purchased from Nation Laboratory Animal Center and CCL2 null mice (B6.129S4-Ccl2 tm1Rol/J) were purchased from the Jackson laboratory. Eight-week-old female mice whose hairs were kept in physiological telogen phase and body weight were around 20–25 g were randomized into different group for experiments. Mice with any focal anagen skin were excluded from the experiments. These mice were maintained by staff at the Animal Facility of Tapiei Veterans General Hospital. These animals were housed in cages

(BCU-2 system, Allentown) with Bed-o'Cobs® 1/4″ bedding (The Andersons, Inc) at 23 ℃, 50 ± 10% humidity on a 12-h light/12-h dark cycle. Mice were fed with Laboratory Autoclavable Rodent Diet 5010® (LabDiet). Food and water were provided without restriction. The animal experiments were compliant with all relevant ethical regulations for animal testing and research and under the approval by the animal care and use committee of Taipei Veterans General Hospital (IACUC Approval No. 2015-038). A total of six independent biological replicates was performed for each group to confirm the findings.

To synchronize the hair cycle, we depilated the 8-week-old mice by waxing. Briefly, melted wax was sprinkled on mice back skin and hairs were stripped away when wax was solidified. Anagen was initiated simultaneously when all the back skin hairs were stripped away. After full anagen development, the consecutive stages (catagen and telogen) were then entered spontaneously in a fairly homogeneous manner. We then performed our stretching experiments on the back skin during the synchronized telogen phase (28 days after depilation).

For macrophage ablation, Clodrosome® (Liposomal Clodronate, Encapsula NanoSciences) or control liposomes were injected subcutaneously (0.05 mg/g) every other day from 4 days before to 4 days after stretching device was fixed. For M2 macrophage ablation, m-Clodrosome® (Mannosylated Clodronate Liposome,

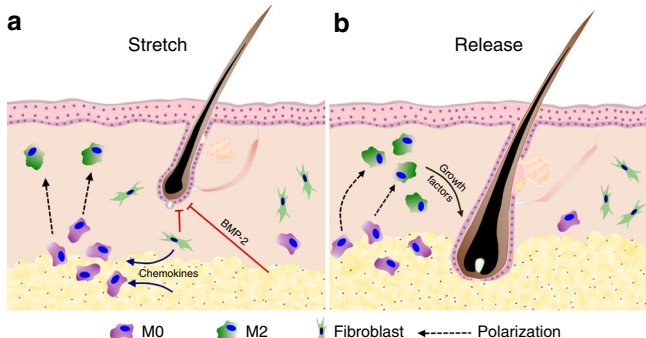

**Fig. 7** Schematic of the molecular basis of stretch-induced hair regeneration. **a** Mechanical stretch promotes BMP-2 production by activating fibroblasts and adipocytes, which inhibit hair regeneration; however, mechanical stretch also stimulates chemokine production to recruit macrophages into the stretched skin. The recruited macrophages undergo M2 polarization in response to a stretched macroenvironment. **b** Macrophages continue to undergo M2 polarization after stretch release and produce growth factors to facilitate anagen initiation

Encapsula NanoSciences) or control liposomes were injected subcutaneously (0.05 mg/g) at day 3, day 5, day 7, and day 10 after stretching. For growth factors injection, beads coated with 7.5 μg HGF (2207-HG, R&D) or IGF-1 (791-MG-050, R&D) were injected subcutaneously into back skin of mice during telogen phase.

**In vivo stretching device**. In vivo skin-stretching device was designed to create programmable stretch. There were 15 checks on the bilateral sides of device with each check measured 1 mm (Supplementary Fig. 1a). We used these checks to quantify the degree of skin elongation, as well as the strain (e.g., 5/15 checks equal to 33% strain). Glue was used to fix the device at proximal and distal sites (Supplementary Fig. 1b) and the mice were taped around to prevent the detachment of stretching device (Supplementary Fig. 1c). Different initial strain was applied as indicated in the results section with increased strain rate (1/15 check = 6.67%) per day to compensate skin cells proliferation which gives rise to decreased strain due to mechano-biologic effects[55].

**Sample collection**. As for evaluating the mechanisms of stretch-induced hair regeneration, we harvested the skin without device-attached area to avoid the interference from hair plucking. The device-attached skin can be clearly recognized and discarded when we collected samples. The sample collectors were different from the operator who processed animal procedures, and thus were blinded to the group allocation.

**Reagents**. The following antibodies were used for immunostaining. Anti-Arginase-1 (Santa cruz, sc-20150, 1:200), anti-BMP-2 (Abcam, ab6285, 1:100 or Bioss, bs-1012R, 1:100), anti-β-catenin (Abcam, ab2365, 1:100), anti-caspase-3 (Epitomics, 1476−1, 1:100), anti-F4/80 (Abcam, ab6640, 1:100), anti- HSP47 (Novus, NBP1-97491, 1:50), anti-K15 (Abcam, ab52816 or ThermoFisher, MA5-11344, 1:100), anti-Ki67 (ThermoFisher, RM-9106, 1:100), anti-perilipin-1 (Abcam, ab3526, 1:100), and anti-phospho-Smad1/5 (Cell signaling, 9516, 1:50).

The following antibodies were used for flow cytometry or cell sorting. Anti-CD45 (BD pharmingen, 553081, 1:200), anti-CD86 (eBioscience, 17-0862, 1:400), anti-CD206 (BioLegend, 141706, 1:40), and anti-F4/80 (eBioscience, 17-4801, 1:10). The corresponding isotype controls were APC/Cy7 Mouse IgG2a, κ (BioLegend, 400230), FITC Rat IgG2a, κ (BD pharmingen, 557228), PE Rat IgG2a, κ (BD pharmingen, 557229), and APC Rat IgG2a, κ (BD pharmingen, 551442).

**Immunostaining**. For immunofluorescence analysis, the collected skin was dissected and fixed with 4% paraformaldehyde in phosphate-buffered saline (PBS), dehydrated, embedded in paraffin, and sectioned at 5 μm. Staining was performed with appropriate antibodies followed by secondary antibodies conjugated to Alexa-488 (ThermoFisher, A-11001, A-11006, or A-11008), Alexa-568 (ThermoFisher, A-11004, A-11011 or A-11077) or Alexa-647 (ThermoFisher, A-21235, A-21244, or A-21247). Quantification of F4/80+ cells in the immunofluorescence study was calculated in 200x field. Quantification of Caspase-3+ cells in the immunofluorescence study was calculated in the field of 200 × 200 μm. Confocal images were obtained with a LSM 880 with AirtScan (ZEISS).

**RT-PCR**. For RNA extraction and RT-PCR, the collected skin samples were cut into pieces and RNA were isolated with miRNeasy Mini kit (QIAGEN) following the manufacturer's recommendations. Transcription levels of target genes were measured by qPCR with TaqMan Fast Universal PCR Master Mix (2×) (Applied

Biosystems, USA). Target gene-specific primer sequences and suitable probes were designed by the Universal Probe Library System software. Relative expression was normalized to GAPDH for each sample. The primers for RT-PCR amplification are summarized in Supplementary Table 3.

**RNA isolation and library preparation for RNA-seq**. The whole layer 5-day stretched skin and non-stretched skin were harvested for RNA-seq. RNAZol kit (Sigma-Aldrich) was used to extract total mRNA. The concentration of RNA was measured by Nanodrop spectrophotometer. Integrity of total RNA was assessed using Agilent 2200 TapeStation-RNA R6K assay (Agilent) and the high quality of RNA (RNA Integrity Number, RIN ≥ 7.5) were used. RNA-seq cDNA libraries were constructed following the standard Illumina protocol. The cDNA libraries were quantitatively measured by qRT-PCR (Roche LightCycler® 480 system) and Qubit Fluorometer (Invitrogen, Carlsbad, California, USA). The cDNA libraries were pooled and sequenced on an Illumina NextSeq 500 platform in single-ended with 76 bps.

**Bioinformatics analysis for RNA-seq data**. Quality control of the RNA-seq reads through FASTX-Toolkit version 0.0.13.2 with high-quality base calling (Phred quality score ≥ 20) and RNA reads longer than 35 bases were retained. Reads were then mapped to the UCSC[56] mouse genome mm10 reference genome using TopHat2[57] version 2.1.0 with option --b2-very-sensitive and -G. The transcripts were assembled, estimated abundances, and normalized by Cufflinks version 2.2.1[58]. Genes with low FPKM value, in both of the compared samples that multiplied FPKM value ≤ 10 ($\log_{10}$ EV ≤ 1), were filtered. Genes were identified as upregulated or downregulated gene if the fold change was greater than or equal to 2 or less than or equal to 0.5 ($\log_2$ FC ≥ 1 or $\log_2$ FC ≤ −1). To understand the function of the differentially expressed genes, functional annotation tools, DAVID[59] (version 6.7), were used to illustrate the biological regulation role of the upregulated or downregulated genes.

**Flow cytometry**. Collected skin samples were immersed in collagenase at 37 °C for 2h and filtered by 70 and 40-μm strainers. Cell suspensions were incubated with the appropriate antibodies for 30 min on ice. FACS Canto II and FACSAria sorters were used for flow cytometry and cell isolation, respectively. All flow cytometry data were analyzed by FlowJo 7.6. The representative gating strategies were shown in Supplementary Fig. 5.

**Macrophage transplantation**. A crude population of BMDMs was collected from mouse bone marrow. Cells were cultured in RPMI 1640 medium containing 10% FBS, 1% penicillin G, and 20 ng/mL recombinant mouse granulocyte-macrophage colony-stimulating factor (R&D, 415-ML/CF,). The medium was replaced on day 2 and 4 and non-adherent cells were rinsed away. Cells were obtained on day 6 as a macrophage population characterized by morphology and high expression of F4/80. For M2 polarization, 10 ng/ml IL-4 (R&D, 404-ML/CF) was added for 24 h before cell harvest. For M1 polarization, 45 ng/ml IFN-γ (BioLegend, #575302) was added for 6 h followed by 100 ng/ml lipopolysaccharides (LPS) (Sigma, L2880) for 6h. Cells were then incubated for 24 h before harvest.

For BMDMs transplantation, polarized or unpolarized macrophages were injected subcutaneously into back skin during the telogen phase. The numbers of transplanted BMDMs were as indicated in the results section.

**Statistical analysis**. All statistical analyses were performed using SPSS 22.0 (SPSS Inc, Chicago, IL, USA) and statistical significance was determined using Student's two-tailed $t$-test or ANOVA followed by a Bonferroni post hoc test (*$P < 0.05$, **$P < 0.01$, ***$P < 0.001$). Each experiment was performed at least three times and $n$ equals the number of independent biological replicates.

**Reporting Summary**. Further information on experimental design is available in the Nature Research Reporting Summary linked to this article.

## Data availability
Data have been deposited in the Gene Expression Omnibus under accession code GSE125806. All other relevant data are available from the corresponding authors upon reasonable request. The source data underlying Figs. 2a–d, 4e, g, 5b–f, 6a, b, and Supplementary Fig. 3b are provided as a Source Data file.

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

## Acknowledgements

The authors acknowledge financial support from the Ministry of Science and Technology (MOST 104-2314- B-075- 054-MY3, 105-2628-B-010-016-MY3, 106-2319-B-400-001, 106-2633-B-010-001, and 107-2314-B-010-015-MY3), National Health Research Institute (NHRI-EX107-10625EI) and Taipei Veterans General Hospital, Taiwan (VN104-12, VN106-13, V104C-055, V105C-033, V106C-030, V106D25-002-MY3-1, V106D25-002-MY3-2, VGHUST104-G1-1-1, VGHUST105-G1-4-1). This work was also partially supported by the Center for Intelligent Drug Systems and Smart Bio-devices (IDS2B) from The Featured Areas Research Center Program within the framework of the Higher Education Sprout Project by the Ministry of Education (MOE) in Taiwan. We thank Professor Nien-Jung Chen for discussion and technical assistance with flow cytometry, and RNA extraction facility in Division of Experimental Surgery, Taipei Veterans General Hospital. This study was also supported by Aiming for the Top University Plan, a grant from Ministry of Education.

## Author contributions

S.Y.C., C.C.C., and Y.T.C. designed the project. S.Y.C., L.Y.L., Y.R.C., S.X.N., P.H.C., P.Y.C., and Y.H.W. performed the experiments and analyzed the data. M.H.Y. designed the in vivo stretching device. C.H.C., H.D.H., Y.C.L., H.M.C., and F.M.L. analyzed the RNA-seq data. C.H.C. and H.D.H. conducted bioinformatics analyses. H.C.H. performed the RNA-seq experiment. OK.L. and C.C.C. supervised the works. S.Y.C., C.C.C., and O.K.L. wrote the manuscript.

## Additional information

**Competing interests:** The authors declare no competing interests.

