## [Peer Review File · Nature Communications]

Editorial Note: Parts of this peer review file have been redacted as indicated to remove third-party material where no permission to publish could be obtained.

Reviewers' comments:

Reviewer #1 (Remarks to the Author):

In their manuscript "Mechanical stretch-induced hair regeneration is "strain" and "duration" dependent", Chu and colleagues examine the contribution of mechanically pulling the skin to of hair follicle cycling. Toward this end, they find robust activating of the hair cycle and associated macrophage recruitment upon stretching. Depletion of macrophages curbs this response and the authors go on to analyze the types of macrophages that contribute to this stretch response and conclude that alternatively active M2 macrophages are key drivers of this response.

While the observations are intriguing, I am concerned about the model of stretch used, the molecular and cellular characterization, with specific focus on lack of controls and the premature conclusions drawn. Thus, the manuscript as it stands is not well-suited for publication in Nature Communications.

Please find below my comments in no particular order of importance:

- Why is the response to strain removal studied (7-9d)? There is no physiologic rationale for this type of input, although it is presumably critical to their hypothesized molecular mechanism, i.e. positive strain induces both Wnt and BMP with similar kinetics and magnitudes, but strain removal is required for BMP downregulation. How do the authors interpret these data?
- Is it unclear if the device used to stretch the mice also induces tissue damage upon attachment? If this is the case, the authors must control for or discern the effects of tissue damage from the stretch response they claim to study
- The authors state parameters for skin strain based seemingly on measurements for deformations of their superficial device. How have the authors actually measured tissue strain? Is there a viscous (time-dependent) component to tissue elongation? The authors cite articles describing active tissue growth to relax the input of tissue strain. Was this measured?
- The use of methods to monitor the Wnt and BMP pathways (RT-PCR for beta-catenin and BMP-2, respectively) is not convincing.
- The analysis of M2 macrophages is rather superficial and also lacks critical controls. For instance, the FACS analysis in Figure 5a of CD206 and CD86 is lacking isotype or FMO controls, making their gating strategy difficult to understand (are a majority of the macrophages not M1 or M2?). Equally, concerning is that the authors did not include CD45 or DAPI in their stain which are critical to include in immune cell analyses within extra lymphoid tissues given their high background. Additionally more than one marker for characterizing M1 and M2 macrophages should be used before coming to this conclusion (the Arg staining in the figure below is too difficult to discern)
- The authors state that "Chemotactic signals for macrophage recruitment come from the dermis and adipose layer, but not epidermis", however this statement is not supported by any functional data. Furthermore, the one chemokine that is functionally tested (in a compartment non-specific manner) has no effect.
- Many of the figures legends and methods are missing crucial details to interpret experiments. For instance, Fig2 the RNAseq- was this on whole skin, if so was it distal to the stretch device?
- In figure 5f the authors do not inject M1 macrophages as a control but rather unpolarized macrophages. This is a critical control for this experiment.
- It seems likely that mechanical stress-induced tissue damage, and not physiologic mechanotransduction pathways, are responsible for their observations. This alternative explanation should be ruled out (or in) through quantification of cell death and other cell stress response pathways (i.e. integrated stress response).

Reviewer #2 (Remarks to the Author):

General comments

In this study Chu et al., tested the hypothesis of how mechanical force affect hair regeneration and induces hair growth in mice. The authors demonstrate that applying tension on the skin modifies hair growth in mice, by increasing M2 polarized macrophage infiltration. In addition, it was also shown that M2 macrophage releases growth factors (HGF, IFG-1, FGF-2) that seem to be responsible for proliferation of hair stem cells. This manuscript contains some novel and potentially clinical relevant data. However, it contains some critical problems, both in experimental design and in data analysis that preclude its recommendation for publication.

Specific comments

1. My major concern about the manuscript is regarding the experimental design of the experiments and data analysis. More recently, there is a growing number of literatures, notably in high impact journals, about the lack of reproducibility of pre-clinical studies. This has led a substantial change in the guidelines for manuscript submission in most relevant international journals, including the Nature group. The present manuscript is not in compliance with such recommendation. For instance, the authors must provide details on the animals used: species, strain, gender, age and body weight. Please also provide general information about the KO mice, especially regarding the genetic background, housing and husbandry conditions (type of cage, bedding material, breeding programme, type of food, access to food and water, environmental enrichment, etc). In my view, the most critical point of the manuscript is how was the sample size calculated? I do not believe that 3 animals per group is sufficient to make any conclusion because there is no statically power nor biological relevance. Are the authors convinced that the number of animals used in this study is sufficient to confirm that the observed statistical differences have biological relevance? Also relevant, to avoid any bias of data interpretation, it is important to provide if the experiments were conducted in a blinded manner (if the experimenter conducted the experiments without prior knowledge of treatments and mouse groups). Were the animals assigned randomly to the various experimental groups? If so, please provide the method of randomization used. These points raised above make this data difficult to be reproduced and weakens the evidence provided by the authors to support the results and conclusions. Indeed, many scientific journals have pointed out the importance of methodological design and have made its details mandatory for publication. Please, see the following references for more details: Landis et al ; Nature 490, 187, 2012; Glenn and Ellis, Nature, 483, 531, 2012; Peers et al., Nature Review Drug Discovery, 11, 273,2012; Collins et al., Nature, 505, 612, 2014; Allison et al., Nature, 530, 27, 2016; Nature, 542, 409, 2017; Science 355, 234, 2017.

2. Furthermore, authors must check whether the statistical analysis used to assess the data (t test) is the most appropriated test. This point together with several aspects raised in the item 1 are extremely critical to sustain the conclusions of the paper.

3. The method section is too much condensed and fails to describe clearly the used protocols. How was the hair cycle synchronization performed? The authors affirm that the animal's hair are in the Telogen phase, but the images in figure 1 (a) and (b) show that animals were only shaved.

4. Why the authors did not use naïve (without device) and sham (with device and without strain) animals in the experiments? I believe that the use of such animals is critical to properly support this experiment.

5. In this manuscript the authors affirmed that mechanical stretch induces hair regeneration, but the experimental model used in this work did not cause any degeneration of the skin or hair follicles. The animals were shaved, a process that do not induce hair cycle alteration neither degeneration of the skin, keeping its physiological conditions.

6. Why do the authors did not present the results of the activation of the hair steam cells, but instead only the presence of these cells in the tissue?

7. The authors should inform the number of the ethic committee approval for this protocol study.
8. How do the authors can correlate the tension applied in the skin with the tension applied in the in vitro fibroblasts tests? There is some previous defined criterion to support this comparison?
9. How were the tensions (20, 33 and 40%) used in the experiments chosen? Is it based in previous experiments?
10. Did the authors shave the animals again in the days 14 and 21? In the figure 1a and 1b, animals seem to be shaved again in the region around the device. Please, clarify this point.
11. Are the images representative of the same animal in each group in the different time points?
12. Authors must inform in the methods section additional data about the used antibodies, notably whether or not their specificities for the proposed target were assessed.

Reviewer #3 (Remarks to the Author):

This is a very interesting manuscript showing that stretch induced hair regeneration through increased M2 macrophages. Strain stimulated cytokines from dermis and adipose layer to recruit macrophages and polarize them into M2 subtypes. Macrophages released various growth factors to induce hair growth. The concept is novel and interesting. I consider this manuscript will be suitable to be published in Nature Communication. There are several questions and I hope the authors can address them.

In Fig. 1a and b, did Day 0 mean the last day when strain was applied for indicated time period or the first day when strain was applied?

(Based on the rest of the paper, I assume Day 0 is the first day when strain is applied.)

When the strain was less than 20%, hair regeneration was not induced. The authors claimed increased M2 macrophages are responsible for the regeneration. Have the authors examined if the number of M2 macrophages increased in 33% strain compared to 20% strain?

33% strain for short term (5days) or long term (10days) negatively affected hair regeneration. Is this also dependent on the number of M2 macrophages?

In Fig. 4, CCL2, CCL3, and CCL12 were up-regulated in dermis and adipose layer to recruit macrophages in stretched skin. The recruited macrophages are supposed to be polarized to become M2 subtype. Did stimuli for M2 polarization like IL4 increase in the stretched skin as well?

In Fig. 5b, the number of M2 macrophages increased gradually while strain was applied, whereas the expression of growth factors such as HGF, IGF-1, FGF-2 and FGF-10 in macrophages suddenly increased at D9 after strain was released. What will be a mechanism to up-regulate the expression of the growth factors in macrophages after the strain is released?

In Fig. 6, they sorted out F4/80+ macrophages that include unpolarized and M1 macrophages. It would be more definitive if they sort out M2 macrophages and examine what growth factors increase in them compared to unpolarized or M1 macrophages.

Response to Reviewer #1:

In their manuscript “Mechanical stretch-induced hair regeneration is “strain” and “duration” dependent”, Chu and colleagues examine the contribution of mechanically pulling the skin to of hair follicle cycling. Toward this end, they find robust activating of the hair cycle and associated macrophage recruitment upon stretching. Depletion of macrophages curbs this response and the authors the go on to analyze the types of macrophages that contribute to this stretch response and conclude that alternatively active M2 macrophages are key drivers of this response.

While the observations are intriguing, I am concerned about the model of stretch used, the molecular and cellular characterization, with specific focus on lack of controls and the premature conclusions drawn. Thus, the manuscript as it stands is not well-suited for publication in Nature Communications.

Answer:

We thank the Reviewer for the comments. We standardized our stretch model and the results were reproducible among our 6 independent replicates in all stretching

condition. We have strengthened the molecular characterization by evaluating
multiple participants in WNT and BMP signaling pathway as well as their
downstream responders (β -catenin, Lef1 and pSmad1/5). We characterized M2
macrophages by multiple biomarkers through flow cytometry
($CD45^+F4/80^+CD206^+CD86^-$), qPCR (Arginase1 and Ym1) and immunostaining
(Arginase1). We also performed the naïve (without device) and sham (with device
under 0% strain) controls in which hair regeneration did not occur, indicating
stretching itself indeed stimulates hair regeneration.

1. Why is the response to strain removal studied (7-9d)? There is no physiologic
rational for this type of input, although it is presumably critical to their hypothesized
molecular mechanism, i.e. positive strain induces both Wnt and BMP with similar
kinetics and magnitudes, but strain removal is required for BMP downregulation.
How do the authors interpret these data?

Answer:

Our previous study had demonstrated that topological hair plucking can induce
threshold dependent efficient regeneration response (Chen et al., Cell 161, 277-290,
2015), thus we hypothesize that maybe “stretch” itself (but not plucking out the hairs)
is enough to trigger hair regeneration response. Based on this, we design the skin
stretching device to evaluate if hairs can regenerate under different “stretch duration”.
In the beginning, we stretched the skin for 1, 3, 5, and 7 days and try to identify the
proper duration which can induce hair regeneration. After testing for more than 10
61 times, we found a 7-day duration leads to hair regeneration, which implied longer
stretch duration may positively affect hair regeneration. Accordingly, we extended the
strain duration to 10 days to see if hair regeneration could occur earlier or even more
luxuriant. Surprisingly, hairs only regenerated in peripheral area in 10-day duration
group as compared to the whole stretched area in 7-day duration group (Fig. 1b and
Supplementary Fig. 2c). These results gave us clues that a too long strain duration
negatively impacted hair regeneration thus strain removal might be as important as
strain given. As a result, we evaluated the biological changes between 7 and 9 days
since it might be the most critical points in hair regeneration in our model.

To probe the possible mechanisms of this intriguing results, we investigated the
alteration of WNT and BMP pathways, as they are major hair cycle activator and
inhibitor, respectively. WNT signals were increased in response to both strain and
strain removal while BMP-2 was upregulated in response to strain but downregulated
when strain released (Fig. 2a-d). We performed immunostaining and found the WNT
signals were mainly in hair matrix (Fig. 2e) while BMP was derived from dermal
fibroblasts (Fig. 2g). In line with our results, Plikus et al., demonstrated the

importance of “intra-follicular” WNT signals and “extra-follicular” BMPs in hair
cycle regulation (Fig. A) (Plikus et al., Nature 451: 340-344, 2008; Plikus et al.,
Science 332: 586-589, 2011). Collectively, as our works were performed on living
organism which is composed of multi-cell population, we think the discrepancy
between WNT and BMPs is because different cell types respond to stretch and stretch
removal disparately. While the main sources of WNT signals come from epithelial
tissue, the BMPs are derived from dermal fibroblasts.

**Fig. A**

[Redacted]

Nature. 2008 Jan 17; 451(7176): 340–344.

2. Is it unclear if the devise used to stretch the mice also induces tissue damage upon
attachment? If this is the case, the authors must control for or discern the effects of
tissue damage from the stretch response they claim to study.

Answer:

We thank the Reviewer for the comment. To exclude the possible influence from
device attachment or removal, we added a group of sham controls (with device and
without strain). Importantly, hair regeneration does not occur in this 0% strain group
(Fig. 1a).

Regarding the influence of tissue damage upon attachment, our previous works
have demonstrated hair plucking leads to hair regeneration (Chen et al., Cell 161, 277-
290, 2015). Similarly, when we used stretching device in this study, hairs were
plucked in device-attached site when stretching device was removed and hair
regeneration was induced in this area (Fig. 1a, showing clearly with * on day 21 under
0 % and 20% strain). However, this kind of hair regeneration is plucking-induced
rather than stretch-induced. Therefore, 0% and 20% strain was determined as non-
regeneration parameter. In contrast, under 33% and 40% strain, hair regeneration
occurred in the whole stretching area in addition to device-attached site (Fig. 1a).
Accordingly, we think hair regeneration under these two parameters (33% and 40%)
are actually stretch-induced rather than plucking-induced as compared to 0% and 20%
strain group.

As for the degree of tissue damage, we have arranged immunostaining of caspase-3
to evaluate the percentage of cell death. There were only scarce apoptotic cells in
stretched skin (Supplementary Fig. 3).

As for evaluating the mechanisms of stretch-induced hair regeneration, we
harvested the skin without device-attached area to avoid the interference from
plucking. The device-attached skin can be clearly recognized and discarded when we
collected samples on day 1, 7, 9, and 14 (Fig. B, arrows: device-attached sites).

3. The authors state parameters for skin strain based seemingly on measurements for
deformations of their superficial device. How have the authors actually measured
tissue strain? Is there a viscous (time-dependent) component to tissue elongation? The
authors cite articles describing active tissue growth to relax the input of tissue strain.
Was this measured?

Answer:

Thank you for your suggestions. We measured the length of tissue within stretching
area on day 1, day 4, and day 7 and found the tissue elongation was really time
dependent. The average lengths on day 1, day 4, and day 7 are 1.73 cm, 1.92 cm, and
2.08 cm, respectively (Fig. C, $n = 3$ for each group, $**p < 0.01$).

The length within stretching area was changing every day and this length can only
be measured after stretching device was removed. Therefore, it's not practical to
remove the device and re-fix it every day (too much damage on attached site.) This is
why we used the strain measured by the device rather than the skin.

4. The use of methods to monitor the Wnt and BMP pathways (RT-PCT for beta-
catenin and BMP-2, respectively) is not convincing.

Answer:

To evaluate the roles of WNTs after skin stretching, we arranged RT-PCR for
*Wnt7b*, *Wnt10a*, as well as *Lef1* which is a nuclear responder of WNT signals (Fig.
2a-c). Both *Wnt7b* and *Lef1* were significantly increased upon stretching and further
upregulated after released. By contrast, *Wnt10a* was peaked during stretching. We
also performed immunostaining for β -catenin which revealed nuclear staining in hair
matrix (Fig. 2e). Collectively, the strain alteration upregulated WNT signaling by
which hair follicles were responded.

We also arranged RT-PCR and immunostaining to elucidate the expression pattern
of BMP signals. Notably, *BMP2* was upregulated upon stretching and significantly
downregulated after strain removal (Fig. 2d). Immunostaining showed the co-
localization of HSP47 (fibroblast marker) and BMP-2 signals (Fig. 2g), implying the
stretch-activated BMP was mainly from dermal fibroblasts. Further, the nuclear
staining of pSmad1/5 within hair stem cells (Fig. 2f) provided evidence that hair stem
cells were inhibited by extra-follicular BMP signals. Taken together, these results
indicated that stretch stimulated fibroblasts to produce BMP-2 which prohibited hair
stem cells from activation.

5. The analysis of M2 macrophages is rather superficial and also lacks critical
controls. For instance, the FACS analysis in Figure 5a of CD206 and CD86 is lacking
isotype or FMO controls, making their gating strategy difficult to understand (are a
majority of the macrophages not M1 or M2?). Equally, concerning is that the authors
did not include CD45 or DAPI in their stain which are critical to include in immune
cell analyses within extra lymphoid tissues given their high background. Additionally
more than one marker for characterizing M1 and M2 macrophages should be used
before coming to this conclusion (the Arg staining in the figure below is too difficult
to discern)

Answer:

We thank the Reviewer for the critical comment. To further ascertain the findings of
M2 increment in our model, CD45 was included for macrophage gating. The
CD45⁺F4/80⁺ CD206⁺CD86⁻ cells (M2 macrophages) were indeed increased in
response to stretch and stretch released (Fig 5a, b). To better characterize the existence
of M2 macrophages, we sorted CD45⁺F4/80⁺ macrophages and arranged qPCR for
*Arginase-1* and *Ym1* (M2 markers). The levels of *Arginase-1* and *Ym1* were
significantly increased in response to both stretch (day 7) and stretch released (day 9)
as compared to non-stretched skin (Fig. 5c, d). Accordingly, these results provided

compelling evidence that macrophage subtype was skewed toward M2 phenotype by
alteration of strain, either increased or decreased.

As for controls of flow cytometry, the figures with isotype control were provided in
Supplementary Fig. 5.

6. The authors state that “Chemotactic signals for macrophage recruitment come from
the dermis and adipose layer, but not epidermis”, however this statement is not
supported by any functional data. Furthermore, the one chemokine that is functionally
tested (in a compartment non-specific manner) has no effect.

Answer:

To evaluate the functional roles of chemokine, we constituted pan-chemokine
inhibitor NR58-3.14.3 (Reckless et al., Immunology 103(2):244-54, 2001) to abolish
the recruitment of macrophages. Nevertheless, the mechanical stretch seems to be too
strong for various chemokine activation (Fig. 4g) that cannot be totally occluded by
NR58-3.14.3 thus there were still heavily infiltrated of macrophages after stretching
(Fig. D, Scale bar = 100µm). Also, we cannot specifically block all kinds of
chemokines within either epidermis, dermis or adipose layer. We now modify our
description to “Mechanical stretch stimulates multiple chemotactic signals that
facilitates macrophage recruitment” in Line 181 of the revised manuscript.

**Fig. D**

Form Our Lab.

7. Many of the figures legends and methods are missing crucial details to interpret
experiments. For instance, Fig2 the RNAseq- was this on whole skin, if so was it
distal to the stretch devise?

Answer:

Thank you for your suggestions. We have revised the figure legends and methods
for the critical details. As for Fig 2 (Fig. 3 in revised manuscript), the samples for
RNA-seq were collected from the whole skin layer (epidermis, dermis and adipose
tissue) within stretched area. The device attached site was discarded to avoid the
interference from hair plucking.

8. In figure 5f the authors do not inject M1 macrophages as a control but rather
unpolarized macrophages. This is a critical control for this experiment.

Answer:

This is a very insightful suggestion. We have polarized macrophages to M1 subtype
by LPS and performed the subcutaneous injection. Hair regeneration does not occur
under different concentration of M1 macrophages injection (1×10^5 , 5×10^5 , 1×10^6 ,
2×10^6 cells in 50 μ l PBS). Now we added these results to our revised Fig. 5j.

9. It seems likely that mechanical stress-induced tissue damage, and not physiologic
mechanotransduction pathways, are responsible for their observations. This
alternative explanation should be ruled out (or in) through quantification of cell death
and other cell stress response pathways (i.e. integrated stress response).

Answer:

This study focuses on the roles of mechanical stretch induced macrophage
recruitment and we think mechanical stress-induced inflammation is responsible for
the hair regeneration. As for the degree of tissue damage, we have performed
immunostaining of caspase-3 to evaluate the percentage of cell death. There were only
scarce apoptotic cells in stretched skin (Supplementary Fig. 3). As for
mechanotransduction pathways, this is an interesting point and may also play roles in
stretch-induced regeneration though it is beyond the main axis of this study. We now
acknowledge this as a possibility and a future direction in the Discussion section. We
will keep evaluating these possibilities in the imminent future. Thank you for your
suggestions.

Response to Reviewer #2:

In this study Chu et al., tested the hypothesis of how mechanical force affect hair
regeneration and induces hair growth in mice. The authors demonstrate that applying
tension on the skin modifies hair growth in mice, by increasing M2 polarized
macrophage infiltration. In addition, it was also shown that M2 macrophage releases
growth factors (HGF, IFG-1, FGF-2) that seem to be responsible for proliferation of
hair stem cells. This manuscript contains some novel and potentially clinical relevant
data. However, it contains some critical problems, both in experimental design and in
data analysis that preclude its recommendation for publication.

Answer:

We thank the Reviewer for the comments. We standardized our experimental
design and increased the sample size to 6 for all of the stretching condition. The
similar phenotype illustrated by more than 6 independent replicates indicated the

novel findings we discovered are reliable. We also performed the naïve (without
device) and sham (with device under 0% strain) controls and hair regeneration did not
occur in both of the control groups. For other details regarding randomization,
blinding, inclusion/exclusion criteria and data analysis, please refer to the responses
for question 1 and 2.

1. My major concern about the manuscript is regarding the experimental design of the
experiments and data analysis. More recently, there is a growing number of
literatures, notably in high impact journals, about the lack of reproducibility of pre-
clinical studies. This has led a substantial change in the guidelines for manuscript
submission in most relevant international journals, including the Nature group. The
present manuscript is not in compliance with such recommendation. For instance, the
authors must provide details on the animals used: species, strain, gender, age and
body weight. Please also provide general information about the KO mice, especially
regarding the genetic background, housing and husbandry conditions (type of cage,
bedding material, breeding programme, type of food, access to food and water,
environmental enrichment, etc). In my view, the most critical point of the manuscript
is how was the sample size calculated? I do not believe that 3 animals per group is
sufficient to make any conclusion because there is no statically power nor biological
relevance. Are the authors convinced that the number of animals used in this study is
sufficient to confirm that the observed statistical differences have biological
relevance? Also relevant, to avoid any bias of data interpretation, it is important to
provide if the experiments were conducted in a blinded manner (if the experimenter
conducted the experiments without prior knowledge of treatments and mouse groups).
Were the animals assigned randomly to the various experimental groups? If so, please
provide the method of randomization used. These points raised above make this data
difficult to be reproduced and weakens the evidence provided by the authors to
support the results and conclusions. Indeed, many scientific journals have pointed out
the importance of methodological design and have made its details mandatory for
publication. Please, see the following references for more details: Landis et al ; Nature
490, 187, 2012; Glenn and Ellis, Nature, 483, 531, 2012; Peers et al., Nature Review
Drug Discovery, 11, 273,2012; Collins et al., Nature, 505, 612, 2014; Allison et al.,
Nature, 530, 27, 2016; Nature, 542, 409, 2017; Science 355, 234, 2017.

Answer:

Thank you for the insightful suggestions. We agree with this point. In order to
prove that our findings are not biased, we exhibit our up to 6 duplicate results in the
revised version (Fig. 1a, b and Supplementary Fig. 2). The similar phenotype and
results illustrated by more than 6 separate experiments further confirm the novel

findings we discover are reliable. In addition, the details of animals and their housing
conditions including type of cage, bedding material, breeding program, type of food,
access to food and water, and environmental enrichment are provided in “Animal
procedures” of Method section.

As for randomization, we first shaved the hairs of 8-week old animals. The mice in
this age were in their physiological synchronization telogen phase (for details, please
refer to response for question 3). We examined the whole back skin and only mice
whose skin was entirely in telogen phase (for details regarding how to differentiate
telogen from anagen phase, please refer to response for question 3) were used for the
experiments. Mice with any focal anagen skin were excluded from the experiments.
The following image shows 10 mice whose skin is in telogen phase (Fig. E). It shows
that almost no differences among these mice’s back skin can be identified. These mice
were randomized to different groups.

**Fig. E**

Form Our Lab

As for blinding, the sample collectors were different from the operator who
processed animal procedures thus were blinded to the group allocation.

2. Furthermore, authors must check whether the statistical analysis used to assess the
data (t test) is the most appropriated test. This point together with several aspects
raised in the item 1 are extremely critical to sustain the conclusions of the paper.

Answer:

Thank you for your suggestion. Though we harvested skin samples at several time
points, all the comparison was conducted between two conditions (e.g., “un-stretched
vs stretched”, “stretched vs released”) thus we used Student’s t-test to assess these
continuous variables.

3. The method section is too much condensed and fails to describe clearly the used
protocols. How was the hair cycle synchronization performed? The authors affirm that
the animal’s hair are in the Telogen phase, but the images in figure 1 (a) and (b) show
that animals were only shaved.

Answer:

We apologize that we did not describe it clear in the method portion. Hair
synchronization means all of the hairs are synchronized to the same hair phase (e.g.,
telogen phase). There are two ways to induce hair synchronization. The first one is
based on the physiological hair cycle. Physiologically, the mice hair cycle follows a
rather precise time-scale (Paus et al, *J Invest Dermatol* 113(4):523-32, 1999; Müller-
Röver et al., *J Invest Dermatol* 117, 3-15, 2001) during the first twelve weeks after
birth. The first physiological synchronization of telogen occurs at week 3 after birth
and the 2nd physiological synchronization of telogen starts from week 7 after birth
(Fig. F). The other way is depilation-induced hair follicle cycling procedure. Briefly,
melted wax is sprinkled on mice back skin and hairs are stripped away when wax is
solidified. Anagen initiated simultaneously when all the back skin hairs were stripped
away. After full anagen development, the consecutive stages (catagen and telogen) are
then entered spontaneously in a fairly homogeneous manner (Paus et al, *J Invest*
*Dermatol* 113(4):523-32, 1999; Müller-Röver et al., *J Invest Dermatol* 117, 3-15,
2001). Because the stripping method will cause some cell damages which may affect
our results, we use physiological synchronization method and choose 8 weeks old
mice as our model to make sure the hairs of the mice are all in the same condition
(telogen phase).

**Fig. F**

[Redacted]

*J. Invest. Dermatol* 117, 3–15,

As for differentiating telogen from anagen phase, we take advantage of the
characteristics of “anagen-melanogenesis coupling” in C57BL/6 mice (Fig. G)
(Müller-Röver et al., *J Invest Dermatol* 117, 3-15, 2001). In other words, melanin is
only produced in skin during anagen phase. Therefore, we can differentiate the
telogen from anagen phase by the color of the skin (pink in telogen and black in
anagen, Fig. H). Further, because the hair shafts are black (due to melanin production
in anagen phase), hair shaving is necessary to observe the skin color and determine
the phases of hair cycle.

Fig. G

[Redacted]

Fig. H

Telogen (pink)

Anagen (black)

Form Our Lab.

J Invest Dermatol 117, 3-15, 2001

In Fig. 1a, b, we shaved the hairs on back skin of mice and performed the study.
Shaving enables us to observe the changes of skin color as well as the alteration of
hair cycles (i.e., telogen or anagen phase) immediately. We can notice the pink color
on day 0 which indicates the study was performed during telogen phase.

4. Why the authors did not use naïve (without device) and sham (with device and
without strain) animals in the experiments? I believe that the use of such animals is
critical to properly support this experiment.

Answer:

Thank you for pointing this out. In fact, we did perform the naïve (without device)
and sham (with device under 0% strain) controls and hair regeneration does not occur
in both control groups. Now we added these control results to our revised Fig. 1a.

5. In this manuscript the authors affirmed that mechanical stretch induces hair
regeneration, but the experimental model used in this work did not cause any
degeneration of the skin or hair follicles. The animals were shaved, a process that do
not induce hair cycle alteration neither degeneration of the skin, keeping its
physiological conditions.

Answer:

Hair follicles undergo cyclic changes of anagen (growth), catagen (regression), and
telogen (quiescence) throughout life. In catagen, hair follicles regress and apoptosis
occurs (physiological degeneration). By contrast, during telogen-to-anagen transition,
the quiescence stem cells are activated and the transit-amplifying daughter cells are

proliferated to form inner root sheath and hair shaft (Fig. I) (Alonso L and Fuchs E. J
Cell Sci 1;119:391-3, 2006). During this process, several lineages of cells are
regenerated (e.g., inner root sheath and outer root sheath). Therefore, the term “hair
regeneration” is extensively used in the field of hair research to represent the process
of hair stem cell activation and telogen-to-anagen transition process (Plikus MV and
Chuong CM. J Invest Dermatol 128(5):1071-80, 2008; Greco et al., Cell Stem Cell
6;4(2):155-69, 2009; Chen et al., Cell 161, 277-290, 2015). In this study, mechanical
stretch induces stem cell activation and telogen-to-anagen transition, thus we affirm
the term of “stretch-induced hair regeneration”.

**Fig. I**

[Redacted]

J Cell Sci 1;119:391-3, 2006

6. Why do the authors did not present the results of the activation of the hair steam
cells, but instead only the presence of these cells in the tissue?

Answer:

K15 is a specific marker of hair stem cells. In order to prove that stretch can
activate hair stem cells and initiate the hair regeneration process, we perform double
staining of K15 and Ki67 (cell proliferation marker) at different time points during the
stretching and releasing period. The double positive staining of K15 and Ki67
demonstrated in Fig. 1d provided direct evidence that hair stem cells were activated
soon after stretch released.

7. The authors should inform the number of the ethic committee approval for this
protocol study.

Answer:

Thank you for your reminding. This study were approved by the Medical Research
Department of Taipei Veterans General Hospital. The number of the ethic committee
approval was provided in “Animal procedures” of Method section. (IACUC Approval
No. 2015-038).

8. How do the authors can correlate the tension applied in the skin with the tension
applied in the in vitro fibroblasts tests? There is some previous defined criterion to
support this comparison?

Answer:

We agree with the Reviewer’s concern that in vitro cell culture cannot totally
represent the in vivo condition although we can use finite-element modeling to
reconstruct the stress map within cells for comparison (Biophys J. 2012 Mar
21;102(6):1303-12). Therefore, we removed the in vitro part and further arranged
immunostaining to better clarify the roles of fibroblasts in the stretch induced hair
regeneration process. The co-localization of HSP47 (fibroblast marker) and BMP-2
signals indicated that the stretching-activated BMP was mainly from dermal
fibroblasts. Further, the nuclear staining of pSmad1/5 within hair stem cells provided
evidence that hair stem cells were inhibited by BMP signals. Now we added these
results to our revised Fig. 2g and 2f.

9. How were the tensions (20, 33 and 40%) used in the experiments chosen? Is it
based in previous experiments?

Answer:

The strain (20, 33, 40%) was chosen based on the device designed. The stretching
device was designed as 15mm x 10mm (Supplementary Fig. 1a). This size covered
the maximal area of back skin that did not restrain the degree of motion of mice after
we fixed the device by glue and tape (Supplementary Fig. 1b, c).

To impose the strain, we divided the 15mm into 15 checks with each check
measured 1mm. Therefore, we can quantify the degree of skin elongation as well as
the strain (e.g., 6/15 checks equal to 40% strain, 5/15 checks equal to 33% strain, and
3/15 checks equal to 20% strain). After fixing the specially-designed stretching device
on the animal, we found 40% strain was the maximal strain that mice skin can endure.
Therefore, we started our experiments from 40% strain and tapered the strain
gradually until stretch-induced hair regeneration did not occur under smaller strain
(3/15 checks = 20% strain).

10. Did the authors shave the animals again in the days 14 and 21? In the figure 1a

and 1b, animals seem to be shaved again in the region around the device. Please,
clarify this point.

Answer:

The hairs “within” stretched area were not shaved through all the experiment
periods. However. The hairs “outside” the stretched area may be shaved if the hair
shafts were getting longer and covered the stretched area which prevented us from
observation the hair cycle within the stretched area.

11. Are the images representative of the same animal in each group in the different
time points?

Answer:

Yes. All of the images of mice represent the same animal in each group.

12. Authors must inform in the methods section additional data about the used
antibodies, notably whether or not their specificities for the proposed target were
assessed.

Answer:

Thank you for reminding us. In the revised version, we provided the details of
antibody in the “Reagents” of Method section.

Response to Reviewer #3:

This is a very interesting manuscript showing that stretch induced hair regeneration
through increased M2 macrophages. Strain stimulated cytokines from dermis and
adipose layer to recruit macrophages and polarize them into M2 subtypes.
Macrophages released various growth factors to induce hair growth. The concept is
novel and interesting. I consider this manuscript will be suitable to be published in
Nature Communication. There are several questions and I hope the authors can
address them.

Answer:

We thank the Reviewer for the compliment. The questions were addressed in detail
as follows.

1. In Fig. 1a and b, did Day 0 mean the last day when strain was applied for indicated
time period or the first day when strain was applied?

(Based on the rest of the paper, I assume Day 0 is the first day when strain is applied.)

Answer:

We thank the Reviewer for pointing this out. The Reviewer is correct that day 0
means the first day when strain is applied. In the revised version, the description has

been added in the figure legend to make it more clearly.

2. When the strain was less than 20%, hair regeneration was not induced. The authors
claimed increased M2 macrophages are responsible for the regeneration. Have the
authors examined if the number of M2 macrophages increased in 33% strain
compared to 20% strain?

Answer:

Thank you for your suggestions. We agreed that this part is very important. To
clarify it, we have arranged the flow cytometry to calculate the numbers of M2
macrophages (CD45⁺F4/80⁺ CD206⁺CD86⁻) under 20% and 33% strain. Strikingly,
the percentage of M2 macrophages was significantly decreased under 20% strain as
compared to 33% strain. This finding suggests that the smaller strain recruits lesser
M2 macrophages which are not sufficient to evoke hair regeneration process. Now we
added these results to our revised Fig. 5f.

3. 33% strain for short term (5days) or long term (10days) negatively affected hair
regeneration. Is this also dependent on the number of M2 macrophages?

Answer:

Thank you for mentioning this interesting point. We arranged flow cytometry to
calculate the numbers of M2 macrophages (CD45⁺F4/80⁺ CD206⁺CD86⁻) in 5-day, 7-
553 day, and 10-day stretching skin. The percentage of M2 macrophages was significantly
decreased in 5-day group as compared to 7-day group. However, there was no
statistically significant differences between 7-day and 10-day group (Fig. J). We think
10-day stretching is a critical borderline condition based on our phenotypic
observation. Among the 6 mice which were stretched for 10 days, total hair
regeneration occurred in 3 mice, and partial (peripheral) hair regeneration occurred in
the other 3. Now these results are provided in our revised Fig. 1b and Supplementary
Figure 2c.

Fig. J

We keep elucidating why hair follicles only regenerate in peripheral area after 10-
572 day stretching. First, we found once hair regeneration occurs (under any stretching
condition), it always begins from the peripheral area (Fig. K) which implies the force
field may not be evenly distributed. Hence, we think the degree of inflammation may
also different in distinct area.

To test this hypothesis, we arranged immunostaining to evaluate the distribution of
F4/80⁺ macrophages after 10-day stretching. Indeed, macrophages are much more
heavily infiltrated in the peripheral area as compared to the central area (Fig. L).
Accordingly, the force received in central area may not be adequate to evoke
inflammatory process that overcome the inhibitory BMP signals under stretching thus
hair regeneration cannot be induced in the central area after 10-day stretching. We
agree that the relationship between force field and inflammation is very interesting
and need further, more definitive experiments to clarify it. Nevertheless, to determine
the relationship between force field and inflammation is far beyond the main axis of
this manuscript. We now acknowledge this as a future direction in the Discussion
section and we will keep working on this issue. Thanks for your question.

4. In Fig. 4, CCL2, CCL3, and CCL12 were up-regulated in dermis and adipose layer
to recruit macrophages in stretched skin. The recruited macrophages are supposed to
be polarized to become M2 subtype. Did stimuli for M2 polarization like IL4 increase
in the stretched skin as well?

Answer:

To address this question, RT-PCR was arranged to evaluate the expression levels of
IL-4 in stretched skin and we did find that IL-4 was rapidly increased on day 1 after
stretching. Now we added this result to our revised Fig. 5e.

5. In Fig. 5b, the number of M2 macrophages increased gradually while strain was
applied, whereas the expression of growth factors such as HGF, IGF-1, FGF-2 and
FGF-10 in macrophages suddenly increased at D9 after strain was released. What will
be a mechanism to up-regulate the expression of the growth factors in macrophages
after the strain is released?

Answer:

We think there might be common upstream transcription factors that response to
mechanical stimulation and also modulate the secretion of various growth factors. We
will keep working on it and acknowledge this as a future direction. Thanks for your
question.

6. In Fig. 6, they sorted out F4/80⁺macrophages that include unpolarized and M1
macrophages. It would be more definitive if they sort out M2 macrophages and
examine what growth factors increase in them compared to unpolarized or M1
macrophages.

Answer:

Thank you for the insightful suggestions. We sorted CD45⁺F4/80⁺CD206⁺ (M2) and
CD45⁺F4/80⁺CD206⁻ (non-M2) cells from 9-day stretched skin and performed RT-
PCR to compare the expression of various growth factors. Notably, HGF and IGF-1
were significantly increased in M2 group as compared to non-M2 group (Fig. 6b).
The functional assay of HGF beads injection further strengthen the roles of growth
factors in hair regeneration (Fig. 6c).

Reviewers' comments:

Reviewer #1 (Remarks to the Author):

In their revised manuscript, Chu and colleagues explore the interesting finding that time and magnitude-dependent mechanical stretch of mouse back skin induces precocious reentry into the hair follicle cycle. The authors have added several new data sets that strengthen the validity of their model and their findings regarding the role of macrophages in stretch induced hair cycling. However, there are still some concerns that should be at the very least textually addressed prior to publication, but really, without experimentally addressing some of the issues below, the paper has notable weaknesses.

- The authors add more biological replicates and a sham control to their main experimental setup, finding the observations sound and repeatable. They also briefly characterize tissue damage and apoptosis in strain versus control groups and saw no differences. Still, many questions remain as to the mechanism of how this phenomenon occurs and how their data can be explained. For example, the finding that a certain strain magnitude (33%) up to 7 days potentiates hair cycling, while strained-time beyond impairs regeneration in the center of the tissue field. This suggests that either prolonged strain is detrimental or conversely that strain-release is beneficial in inducing regeneration. Since this is at the crux of their observation, experiments to explore these hypotheses are really needed. Also, discussion of these topics is warranted.
- The mechanism by which M2 macrophages promote regeneration is overstated based on the data provided. In particular the statement "M2 macrophages facilitate hair regeneration by producing various growth factors" seems premature given the data presented merely observe growth factor expression by M2 macrophages but not functionally test their relative contribution to in vivo.
- Minor point: For FACS data, isotype negative should be overlaid in the flow plots on the main figures.

Reviewer #2 (Remarks to the Author):

The revised version of the manuscript of Chu et al., is improved when compared with the previous one. Authors have answered most points raised by this referee, however most critical points still remain unclear or were not adequately solved.

1. The authors have used 6 animals per group of the stretching condition, however this number of animals per group was observed only in figure 1. In the subsequent experiments the number of animal per group is only 3, which is totally inadequate sample to permit a robust and reliable statistical analysis necessary to support the study conclusion and also to allow the reproducibility of the experiments by independent researchers.
2. In most assays the authors used more than two experimental groups, so the used t-test for statistical analysis is not appropriate. The one-way analysis of variance (ANOVA) followed by a post hoc test must be used.
3. Authors assessed the whole back skin of the animals and selected those who were in the telogen phase for further experiments. How do the authors claim that all animals were in the telogen phase? Based on the previous experience of this reviewer, it is not common to find a total homogeneity of the animal's back skin, indicating that the animal age is not always the best way to select the hair growth phase.
4. In the figure 1D, it does not clear the presence of higher cell proliferation (Ki-67+) in the hair stem cells (K15+). Furthermore, cell proliferation is different from cell activation, since it is possible to observe cell activation through release of some mediators without cell proliferation.

5. In the figure 2F and G, it is quite difficult to observe co-localization between the HSP-47 and BMP-2, which compromises the statement, that stretching-activated BMP-2 mainly from dermal fibroblasts. In addition, the immunofluorescence analysis alone does not support the hypothesis of activation/inhibition of signaling pathways.

Reviewer #3 (Remarks to the Author):

My previous comments have been addressed. I have no additional comments.

Response to Reviewer #1:

In their revised manuscript, Chu and colleagues explore the interesting finding that time and magnitude-dependent mechanical stretch of mouse back skin induces precocious reentry into the hair follicle cycle. The authors have added several new data sets that strengthen the validity of their model and their findings regarding the role of macrophages in stretch induced hair cycling. However, there are still some concerns that should be at the very least textually addressed prior to publication, but really, without experimentally addressing some of the issues below, the paper has notable weaknesses.

1. The authors add more biological replicates and a sham control to their main experimental setup, finding the observations sound and repeatable. They also briefly characterize tissue damage and apoptosis in strain versus control groups and saw no differences. Still, many questions remain as to the mechanism of how this phenomenon occurs and how their data can be explained. For example, the finding that a certain strain magnitude (33%) up to 7 days potentiates hair cycling, while strained-time beyond impairs regeneration in the center of the tissue field. This suggests that either prolonged strain is detrimental or conversely that strain-release is beneficial in inducing regeneration. Since this is at the crux of their observation, experiments to explore these hypotheses are really needed. Also, discussion of these topics is warranted.

Answer:

We thank the Reviewer for the critical comment. One intriguing finding in this study is that, prolonged strain seems to be detrimental in hair regeneration, since 10-day strain duration leads to only partial hair regeneration (Fig. 1b, Supplementary Fig. 2c). To decipher this phenomenon, we first hypothesized that different stretching duration leads to different numbers of M2 macrophages since M2 macrophages play essential roles in stretch-induced hair regeneration. To test this hypothesis, we arranged flow cytometry to calculate the numbers of M2 macrophages (CD45⁺F4/80⁺CD206⁺CD86⁻) in 7-day, and 10-day stretching skin. However, there was no statistically significant difference between 7-day and 10-day group (Supplementary Fig. 4b). Nevertheless, we noticed that once hair regeneration occurs under any stretching condition, it always begins from the peripheral area (Fig. A), implying that the force field may not be evenly distributed. Hence, we think the degree of inflammation may also be different between peripheral and central areas. To test this hypothesis, we arranged immunostaining to evaluate the distribution of F4/80⁺ macrophages. Indeed, macrophages are much more heavily infiltrated in the

peripheral area as compared to the central area after 10-day stretching. In contrast, the macrophages are distributed relatively evenly in 7-day stretching group (Supplementary Fig. 4a). Since we increased strain rate (1 mm/day) every day in the experiment, the skin of 10-day stretching group is thus 3 mm longer than that of the 7-day stretching group. Accordingly, the mechanical stimulus exerted in central area may not be adequate to evoke inflammatory process that induced hair regeneration. Indeed, how different strain conditions alter the hair growth patterns is important and we have added a paragraph to discuss this topic in Discussion section (Line 304-320).

As for the relationship between force field and inflammation, it will be very interesting and require more elucidation. Nevertheless, to determine the relationship between force field and inflammation is beyond the main axis of this study and this will be our future direction. We will keep working on this issue. Thank you for your suggestion.

2. The mechanism by which M2 macrophages promote regeneration is overstated based on the data provided. In particular the statement “M2 macrophages facilitate hair regeneration by producing various growth factors” seems premature given the data presented merely observe growth factor expression by M2 macrophages but not functionally test their relative contribution to in vivo.

Answer:

Thank you for mentioning this point. As *Hgf* and *Igf1* were significantly upregulated in M2 macrophages compared to non-M2 cells (Fig. 6b), we performed in vivo functional analysis of these two factors. Beads coated with HGF or IGF-1 were injected into back skin of mice during telogen phase. Within 5 days after beads implantation, hair stem cells within the hair follicles adjacent to either HGF- or IGF-1-coated beads displayed proliferation by expression of Ki67 (Fig. 6c). These findings provide direct evidence that both HGF and IGF-1 expressed by M2 macrophages may activate hair stem cells.

3. Minor point: For FACS data, isotype negative should be overlaid in the flow plots

on the main figures.

Answer:

Thank you for your reminding. The figures have been revised with isotype overlaid (Fig. 5a).

Reviewer #2 (Remarks to the Author):

The revised version of the manuscript of Chu et al., is improved when compared with the previous one. Authors have answered most points raised by this referee, however most critical points still remain unclear or were not adequately solved.

1. The authors have used 6 animals per group of the stretching condition, however this number of animals per group was observed only in figure 1. In the subsequent experiments the number of animal per group is only 3, which is totally inadequate sample to permit a robust and reliable statistical analysis necessary to support the study conclusion and also to allow the reproducibility of the experiments by independent researchers.

Answer:

Thank you for your reminding. We have performed more experiments per the comments of the Reviewer and have increased the number of animals to 6 in each experimental group accordingly. All of the results are in line with our previous findings except CCL7. The level of CCL7 is significantly increased after 7-day stretching. We have revised the manuscript (Line 195). Thank you for your suggestion.

2. In most assays the authors used more than two experimental groups, so the used t-test for statistical analysis is not appropriate. The one-way analysis of variance (ANOVA) followed by a post hoc test must be used.

Answer:

Thank you for the suggestions. The statistical analysis has been revised with ANOVA followed by a Bonferroni post hoc test (Fig. 2a-d, 4g, 5b-e, 6a and Supplementary Fig. 3b).

3. Authors assessed the whole back skin of the animals and selected those who were in the telogen phase for further experiments. How do the authors claim that all animals were in the telogen phase? Based on the previous experience of this reviewer, it is not common to find a total homogeneity of the animal's back skin, indicating that the animal age is not always the best way to select the hair growth phase.

Answer:

We agree with the reviewer that, indeed, a total homogeneity of the mice's back skin is uncommon if we take account for the "whole lifetime". However, the hair follicles of C57BL/6 mice undergoes a stereotypic cycle during the "first 14 weeks after birth", and thus the exact timing of hair growth (anagen), regression (catagen) and resting (telogen) phases can still be accurately predicted during this period according to previous reported works by other groups (Fig. B) (Rompolas et al., *Nature* 502(7472):513-8, 2013; Müller-Röver et al., *J Invest Dermatol* 117, 3-15, 2001).

Fig. B

[Redacted]

J. Invest. Dermatol 117, 3–15, 2001.

Based on this important feature, there are high impact papers evaluating physiologic characteristics or molecular pathway according to this strict time-scale (Greco et al., *Cell Stem Cell* 4(2):155-69, 2009; Hsu et al., *Cell* 144(1): 92–105, 2011; Festa et al., *Cell* 146(5):761-71, 2011; Oshimori et al., *Cell Stem Cell* 10(1):63–75, 2012; Kandyba et al., *Proc Natl Acad Sci U S A* 110(4):1351-6, 2013; Rompolas et al., *Nature* 502(7472):513-8, 2013).

Nevertheless, in order to eliminate any individual variance, we depilated the mice to synchronize the hair cycle. Briefly, melted wax is sprinkled on mice back skin and hairs are stripped away when wax is solidified. Anagen was initiated simultaneously when all the back skin hairs were stripped away. After full anagen development, the consecutive stages (catagen and telogen) were then entered spontaneously in a fairly homogeneous manner. We then repeated our stretching experiments on their back skin during "synchronized telogen (28 days after depilation)". Notably, hair growth occurs in all of the depilated mice after stretching (n=6, Supplementary Fig. 2d) which provides compelling evidence that mechanical stretch indeed stimulates telogen-to-anagen transition and hair regeneration.

4. In the figure 1D, it does not clear the presence of higher cell proliferation (Ki-67+) in the hair stem cells (K15+). Furthermore, cell proliferation is different from cell activation, since it is possible to observe cell activation through release of some mediators without cell proliferation.

Answer:

Hair stem cells are kept in quiescence in telogen phase (Greco et al., *Cell Stem Cell* 4(2):155-69, 2009). Upon activation, only a few hair stem cell proliferation can fuel the whole telogen-to-anagen transition process (Greco et al., *Cell Stem Cell* 4(2):155-69, 2009). Previous study has demonstrated only one or two Ki67⁺ cells/ follicle can be found during anagen initiation (Fig. C) (Osorio et al., *Development* 135(6):1059-68, 2008). Our immunostaining reveals several Ki67⁺ hair stem cells on day 8 and day 9 (Fig. 1d) and the numbers of proliferated stem cells were in accordance with those in previous researches.

Fig. C

[Redacted]

Development 135(6):1059-68, 2008.

Based on the characteristic of long-term quiescence and rarely proliferation of hair stem cells, proliferation is commonly used as a marker for hair stem cell activation (Osorio et al., *Development* 135(6):1059-68, 2008; Greco et al., *Cell Stem Cell* 4(2):155-69, 2009; Castellana et al., *PLoS Biol* 23;12(12):e1002002, 2014). That being said, we agree with the reviewer that cell proliferation is not equal to cell activation. We have modified our wording in the revised manuscript (Abstract and Figure Legend 1). Thank you for the suggestion.

5. In the figure 2F and G, it is quite difficult to observe co-localization between the HSP-47 and BMP-2, which compromises the statement, that stretching-activated BMP-2 mainly from dermal fibroblasts. In addition, the immunofluorescence analysis alone does not support the hypothesis of activation/inhibition of signaling pathways.

Answer:

Thank you for pointing this out. We have performed the confocal microscopy and revealed that BMP-2 signals were indeed co-localized with HSP-47⁺ cells (Fig. 2i). To evaluate whether other cell types participate in BMP-2 production, we also performed dual immunostaining for BMP-2 and Perilipin-1 (markers of adipocytes) since previous study demonstrated the extrafollicular BMP signals come from both dermis and adipose tissue (Plikus et al., *Nature* 451: 340-344, 2008). Perilipin-1⁺BMP-2⁺ cells can be found in 7-day stretched skin (Fig. 2j, k), implying that in addition to dermal fibroblasts, BMP-2 can also come from adipocytes. We have revised the description in

the manuscript (Line 141). Thank you for the reminder.

In addition to immunostaining, the upregulation of *bmp2* upon stretching was also affirmed by qPCR (Fig. 2d), indicating the role of mechanical stretch in activating BMP-2 signals. Further, as the phosphorylation and nuclear translocation of SMAD1/5/8 is the well-known downstream signals of BMP-2 (Vanhatupa et al., Stem Cells Transl Med 4(12):1391-402, 2015), the expression of nuclear pSMAD1/5 in hair stem cells thus provides evidence that BMP-2 indeed affected hair stem cells (Fig. 2g).

REVIEWERS' COMMENTS:

Reviewer #1 (Remarks to the Author):

Chu and colleagues show that stretching of the skin with a defined magnitude and duration induces hair regeneration. They have addressed some of the reviewer comments adequately. However there are a few troublesome concerns that remain.

-The authors need to be up front in discussing the role of tissue damage (e.g. wound healing) and its well established role in promoting hair cycle reentry. In this regard, it is not all that surprising that stretching also induces hair regeneration. Even though the authors don't see increased cell death after strain application, disruption of tissue components (e.g. ECM) and release of such debris or temporary barrier disruption (e.g. rupture of cell junctions) could offer a rather trivial explanation for the observed macrophage recruitment, polarization, and stimulation of hair cycle activation. This needs to be discussed and the authors need to be transparent about this. It is quite possible that the authors haven't really made much of an advance in their work. Nevertheless, the stretching offers another means of introducing tissue strain, and if properly placed in the context of other work, skin biologists will be interested in these results.

Geometric changes and their effect on morphogen gradients or "direct" force-transmitting/sensing pathways could be operative, which would be of interest in following up on. Although deciphering between such potential mechanisms -- linking force to CCL2, if that is indeed the physiologic recruitment signal in response to stretch -- is not required at this stage, the authors should acknowledge prior work on the role of tissue damage and how it might be operative in their system.

-The authors state "Since we increased strain rate (1 mm/day) every day in the experiment, the skin of 10-day stretching group is thus 3 mm longer than that of the 7-day stretching group. Accordingly, the mechanical stimulus exerted in central area may not be adequate to evoke inflammatory process that induced hair regeneration. Indeed, how different strain conditions alter the hair growth patterns is important and we have added a paragraph to discuss this topic in Discussion section(Line 304-320)." This logic does not hold. If the 10d stretch is greater than 7d, the simplest argument is that force should be higher throughout the tissue (perhaps elevated at the periphery), but not less. This is disregarding the effects of passive strain relaxation and tissue remodeling, which may change stress/strain distributions over relevant time scales. The authors show no compelling data for what the stress/strain distributions are, how they change over time, or how they are specifically different between 7d and 10d. Thus any mention of spatial stress distributions should be removed or else new experiments should be done to substantiate these presently unfounded claims.

-There are numerous language and grammatical mistakes that need to be corrected before publication.

Given the nature of these concerns, the manuscript is not yet appropriate for publication.

Reviewer #2 (Remarks to the Author):

This referee reviewed carefully the latest revised version of the MS submitted by the authors. The authors were able to answer some critical questions raised previously by the reviewer, notably about the necessity to increase the number of animals per group (from 3 to 6). Furthermore they have also

used an appropriated statistical analysis methodology to compare the data. Other minor points raised by the reviewer were also answered and / or altered in the manuscript.

Response to Reviewer #1:

Chu and colleagues show that stretching of the skin with a defined magnitude and duration induces hair regeneration. They have addressed some of the reviewer comments adequately. However there are a few troublesome concerns that remain.

1. The authors need to be up front in discussing the role of tissue damage (e.g. wound healing) and its well established role in promoting hair cycle reentry. In this regard, it is not all that surprising that stretching also induces hair regeneration. Even though the authors don't see increased cell death after strain application, disruption of tissue components (e.g. ECM) and release of such debris or temporary barrier disruption (e.g. rupture of cell junctions) could offer a rather trivial explanation for the observed macrophage recruitment, polarization, and stimulation of hair cycle activation. This needs to be discussed and the authors need to be transparent about this. It is quite possible that the authors haven't really made much of an advance in their work. Nevertheless, the stretching offers another means of introducing tissue strain, and if properly placed in the context of other work, skin biologists will be interested in these results.

Geometric changes and their effect on morphogen gradients or "direct" force-transmitting/sensing pathways could be operative, which would be of interest in following up on. Although deciphering between such potential mechanisms -- linking force to CCL2, if that is indeed the physiologic recruitment signal in response to stretch -- is not required at this stage, the authors should acknowledge prior work on the role of tissue damage and how it might be operative in their system.

Answer:

Thank you for pointing out these important issues. The paragraph discussing the possible roles of tissue damages, including the disruptions of extracellular matrix or rupture of cell junction, has been added in Discussion section (Line 332-340).

We also think the links among force distribution, tissue damage, and morphogen gradients are fields of great interest and we acknowledge these works as future direction in Discussion section (Line 340-342). Thank you for your valuable suggestion.

2. The authors state "Since we increased strain rate (1 mm/day) every day in the experiment, the skin of 10-day stretching group is thus 3 mm longer than that of the 7-day stretching group. Accordingly, the mechanical stimulus exerted in central area may not be adequate to evoke inflammatory process that induced hair regeneration. Indeed, how different strain conditions alter the hair growth patterns is important and

we have added a paragraph to discuss this topic in Discussion section (Line 304-320).”

This logic does not hold. If the 10d stretch is greater than 7d, the simplest argument is that force should be higher throughout the tissue (perhaps elevated at the periphery), but not less. This is disregarding the effects of passive strain relaxation and tissue remodeling, which may change stress/strain distributions over relevant time scales. The authors show no compelling data for what the stress/strain distributions are, how they change over time, or how they are specifically different between 7d and 10d. Thus any mention of spatial stress distributions should be removed or else new experiments should be done to substantiate these presently unfounded claims.

Answer:

We thank the Reviewer for the comment. The paragraph regarding the spatial stress distributions has been removed in Discussion section and how strain distribution attributes to morphogen gradients and downstream cell responses will be our future directions (Line 340-342). Thank you for the valuable suggestion

3. There are numerous language and grammatical mistakes that need to be corrected before publication.

Answer:

The manuscript has been sent for English editing and the proof was shown below.